# Sharp uniform convergence bounds through empirical centralization

**Cyrus Cousins**
Department of Computer Science
Brown University
Providence, RI 02912
ccousins@cs.brown.edu

**Matteo Riondato**
Department of Computer Science
Amherst College
Amherst, MA 01002
mriondato@amherst.edu

## Abstract

We introduce the use of *empirical centralization* to derive novel practical, probabilistic, sample-dependent bounds to the Supremum Deviation (SD) of empirical means of functions in a family from their expectations. Our bounds have optimal dependence on the maximum (i.e., wimpy) variance and the function ranges, and the same dependence on the number of samples as existing SD bounds. To compute the bounds in practice, we develop novel tightly-concentrated Monte-Carlo estimators of the empirical Rademacher average of the empirically-centralized family, and we show novel concentration results for the empirical wimpy variance. Our experimental evaluation shows that our bounds greatly outperform non-centralized bounds and are extremely practical even at small sample sizes.

## 1 Introduction

The *supremum deviation* of the empirical means of functions in a family $\mathcal{F} \subseteq \mathcal{X} \to [a,b] \subset \mathbb{R}$ from their expectations is a key object in the study of empirical processes [23]. Formally, let $\mathcal{D}$ be a distribution on the domain $\mathcal{X}$ and $\boldsymbol{x} = \{x_1, \dots, x_m\}$ be a collection of $m$ independent samples from $\mathcal{D}$. The *Supremum Deviation (SD) of $\mathcal{F}$ on $x$* is the quantity

$$\mathsf{SD}(\mathcal{F}, \boldsymbol{x}) \doteq \sup_{f \in \mathcal{F}} \left| \hat{\mathbb{E}}_{\boldsymbol{x}}[f] - \mathbb{E}_{\mathcal{D}}[f] \right|, \text{ where } \hat{\mathbb{E}}_{\boldsymbol{x}}[f] \doteq \frac{1}{m} \sum_{i=1}^{m} f(x_i) \ .$$

The sample-dependent *Empirical Rademacher Average (ERA)* $\hat{\mathsf{R}}_m(\mathcal{F}, \boldsymbol{x})$ *of $\mathcal{F}$ on $\boldsymbol{x}$* and its expectation, the *Rademacher Average (RA)* $\mathsf{R}_m(\mathcal{F}, \mathcal{D})$ *of $\mathcal{F}$* [3, 13], allow to derive upper and lower bounds to the SD (see (2)). Let $\boldsymbol{\sigma}$ be a collection of $m$ independent Rademacher variables (i.e., uniform on $\{-1, 1\}$). These two quantities are defined as[1]

$$\hat{\mathsf{R}}_m(\mathcal{F}, \boldsymbol{x}) \doteq \mathbb{E}_{\boldsymbol{\sigma}} \left[ \sup_{f \in \mathcal{F}} \left| \frac{1}{m} \sum_{i=1}^{m} \sigma_i f(x_i) \right| \right], \text{ and } \mathsf{R}_m(\mathcal{F}, \mathcal{D}) \doteq \mathbb{E}_{\boldsymbol{x}} \left[ \hat{\mathsf{R}}_m(\mathcal{F}, \boldsymbol{x}) \right] \ . \tag{1}$$

The RA controls the finite-sample *expected* SD as [28]

$$\tfrac{1}{2} \mathsf{R}_m(\mathcal{F}, \mathcal{D}) - \tfrac{1}{\sqrt{m}} \sup_{f \in \mathcal{F}} \|f\|_\infty \leq \mathbb{E}_{\boldsymbol{x}}[\mathsf{SD}(\mathcal{F}, \boldsymbol{x})] \leq 2\mathsf{R}_m(\mathcal{F}, \mathcal{D}) \ . \tag{2}$$

Probabilistic deviation bounds can be obtained by studying the convergence properties of the SD, and sample-dependent versions use the ERA and its deviation from the RA (see

also Thm. 3). The dependence on the *maximum* $q \doteq \sup_{f \in \mathcal{F}} \|f\|_\infty$ of $\mathcal{F}$ makes the lower bound unsatisfactory, as this quantity can be very large. This downside is particularly evident at relatively small sample sizes, which are actually the most interesting in practice. As uniform convergence bounds are now used not "just" for the theoretical analysis of the performance of learning, but also to develop randomized approximation algorithms for many tasks [1, 21, 22, 24, 25], we believe it is extremely important to derive *practical* bounds to the SD that are optimized not just in terms of the number of samples, but also of other important parameters, such as the maximum and the *wimpy variance* (see (8)). In this work, we use various forms of *centralization* to develop such practical bounds. Define the *distributional centralization* $\mathsf{C}_{\mathcal{D}}(\mathcal{F})$ *w.r.t.* $\mathcal{D}$ as the family

$$\mathsf{C}_{\mathcal{D}}(\mathcal{F}) \doteq \{x \mapsto f(x) - \mathbb{E}_{\mathcal{D}}[f], f \in \mathcal{F}\} \ . \tag{3}$$

$\mathsf{C}_{\mathcal{D}}(\mathcal{F})$ contains one function $g$ for each $f \in \mathcal{F}$, such that $g$ is $f$ shifted by its expectation w.r.t. $\mathcal{D}$, thus, $\mathbb{E}_{\mathcal{D}}[g] = 0$ for each $g \in \mathsf{C}_{\mathcal{D}}(\mathcal{F})$. The Rademacher Average $\mathsf{R}_m(\mathsf{C}_{\mathcal{D}}(\mathcal{F}), \mathcal{D})$ of $\mathsf{C}_{\mathcal{D}}(\mathcal{F})$ *sharply* controls the finite-sample expected SD as [7, Lemma 11.4]

$$\frac{1}{2}\mathsf{R}_m(\mathsf{C}_{\mathcal{D}}(\mathcal{F}), \mathcal{D}) \leq \mathbb{E}_{\boldsymbol{x}}[\mathsf{SD}(\mathcal{F}, \boldsymbol{x})] \leq 2\mathsf{R}_m(\mathsf{C}_{\mathcal{D}}(\mathcal{F}), \mathcal{D}) \ . \tag{4}$$

Comparing (2) and (4), it is evident that the RA could be an *arbitrary large* multiplicative factor away from the expected SD, especially at small-sample regimes or when the maximum $q$ of $\mathcal{F}$ is large. The RA of the distributional centralization instead is *always* at most a multiplicative factor *two* away in both directions. Distributional centralization is therefore already known to be beneficial in the *expected* case, but can this gain be generalized to the probabilistic case, possibly using only sample-dependent quantities?

**Contributions.** In this work we introduce the use of *empirical centralization* (see Sect. 2) to derive *practical, probabilistic* bounds to the SD. Our bounds exhibit a better or no worse dependence on important parameters such as the wimpy variance, the range (see (6)), and the sample size $m$ (see Thm. 4). We also show that the dependence on the wimpy variance that we obtain is optimal (Lemma 2 and Coro. 1). We introduce a novel empirical counterpart to the RA of the distributional centralization which uses *empirical centralization* to bound the SD. We analyze the bias of this quantity (Lemma 1) and derive its concentration properties (Thm. 1) using tail bounds for *self-bounding functions* [4, 6]. In order to obtain fully-sample-dependent bounds, we introduce a Monte-Carlo estimation approach with novel tight deviation bounds (Thm. 5), and we also develop novel tight bounds for the empirical wimpy variance (Thm. 2), which we believe to be of independent interest. The results of our experimental evaluation show the advantages of centralization: the computed bounds to the SD are much smaller than those computed without centralization, even at small sample sizes. Due to space restrictions, all our proofs are in the supplementary material.

## 2 Empirical centralization

We define the *empirical centralization* $\hat{\mathsf{C}}_{\boldsymbol{x}}(\mathcal{F})$ of $\mathcal{F}$ *w.r.t. the sample* $\boldsymbol{x} \in \mathcal{X}^m$ as

$$\hat{\mathsf{C}}_{\boldsymbol{x}}(\mathcal{F}) \doteq \{x \mapsto f(x) - \hat{\mathbb{E}}_{\boldsymbol{x}}[f], f \in \mathcal{F}\} \ .$$

This quantity is an empirical counterpart to the distributional centralization $\mathsf{C}_{\mathcal{D}}(\mathcal{F})$ of $\mathcal{F}$ (see (3)). The key quantity that we use to derive the sample-dependent probabilistic bounds to the SD (Sect. 3) is the ERA of the empirical centralization of $\mathcal{F}$, i.e., the quantity

$$\hat{\mathsf{R}}_m(\hat{\mathsf{C}}_{\boldsymbol{x}}(\mathcal{F}), \boldsymbol{x}) \ .$$

This quantity is completely dependent on the realized $\boldsymbol{x}$, even more, in some sense, than a "standard" ERA (see (1)), because the considered family $\hat{\mathsf{C}}_{\boldsymbol{x}}(\mathcal{F})$ is also a function of $\boldsymbol{x}$, i.e., it is *sample-dependent*. We now derive its important properties: bias and concentration.

**Bias** The expectation w.r.t. $\boldsymbol{x}$ of $\hat{\mathsf{R}}_m(\hat{\mathsf{C}}_{\boldsymbol{x}}(\mathcal{F}), \boldsymbol{x})$ is *not* the RA of the distributional centralization of $\mathcal{F}$ (i.e., $\mathsf{R}_m(\mathsf{C}_{\mathcal{D}}(\mathcal{F}), \mathcal{D})$), but we now show that the bias decreases rapidly in $m$, i.e., $\mathsf{R}_m(\mathsf{C}_{\mathcal{D}}(\mathcal{F}), \mathcal{D}) \in \Theta(\mathbb{E}_{\boldsymbol{x}}[\hat{\mathsf{R}}_m(\hat{\mathsf{C}}_{\boldsymbol{x}}(\mathcal{F}), \boldsymbol{x})])$. For ease of notation, let

$$\mathsf{b}(m) \doteq \mathbb{E}_{\boldsymbol{\sigma}}\left[\left|\frac{1}{m}\sum_{i=1}^m \sigma_i\right|\right] \ \left(\text{which is } \Theta\left(\frac{1}{\sqrt{m}}\right)\right) \ . \tag{5}$$

**Lemma 1.** *Suppose $m \geq 4$. Then*

$$\frac{\mathbb{E}_{\boldsymbol{x}}\left[\hat{\mathsf{R}}_m(\hat{\mathsf{C}}_{\boldsymbol{x}}(\mathcal{F}), \boldsymbol{x})\right]}{1 + 2\mathsf{b}(m)} \leq \mathsf{R}_m(\mathsf{C}_{\mathcal{D}}(\mathcal{F}), \mathcal{D}) \leq \frac{\mathbb{E}_{\boldsymbol{x}}\left[\hat{\mathsf{R}}_m(\hat{\mathsf{C}}_{\boldsymbol{x}}(\mathcal{F}), \boldsymbol{x})\right]}{1 - 2\mathsf{b}(m)} \quad .$$

**Concentration** We now show that $\hat{\mathsf{R}}_m(\hat{\mathsf{C}}_{\boldsymbol{x}}(\mathcal{F}), \boldsymbol{x})$ is tightly concentrated around its expectation because it is a *self-bounding function* [4, 6] (see also Def. 2 in the supplementary material). We call the *widest range of $\mathcal{F}$* the quantity

$$r \doteq \sup_{f \in \mathcal{F}}(\max_{x \in \mathcal{X}} f(x) - \min_{y \in \mathcal{X}} f(y)) \quad (\leq b - a) \quad . \tag{6}$$

It is possible that $r \ll b - a$, for example, when $\mathcal{F}$ contains a function $f$ and a function $g = f + c$ for some $c \in \mathbb{R}$. The widest range of the empirical and distributional centralizations of $\mathcal{F}$ is the same as the widest range of $\mathcal{F}$.

**Theorem 1.** *Suppose $m \geq 1$, and let $\chi \doteq 1 + 2\mathsf{b}(m)$. For any $\delta \in (0, 1)$, with probability at least $1 - \delta$ over the choice of $\boldsymbol{x}$, it holds that*

$$\mathbb{E}_{\boldsymbol{x}}[\hat{\mathsf{R}}_m(\hat{\mathsf{C}}_{\boldsymbol{x}}(\mathcal{F}), \boldsymbol{x})] \leq \hat{\mathsf{R}}_m(\hat{\mathsf{C}}_{\boldsymbol{x}}(\mathcal{F}), \boldsymbol{x}) + \frac{2r\chi\ln\frac{1}{\delta}}{3m} + \sqrt{\left(\frac{r\chi\ln\frac{1}{\delta}}{\sqrt{3}m}\right)^2 + \frac{2r\chi(\hat{\mathsf{R}}_m(\hat{\mathsf{C}}_{\boldsymbol{x}}(\mathcal{F}), \boldsymbol{x}) + r\mathsf{b}(m))\ln\frac{1}{\delta}}{m}} \quad . \tag{7}$$

The ERA of $\mathcal{F}$ is a self-bounding function [5, Sect. 5.1], but proving this fact for the ERA of the empirical centralization $\hat{\mathsf{C}}_{\boldsymbol{x}}(\mathcal{F})$ of $\mathcal{F}$ is more challenging (see proof in the supplementary material), because the empirical centralization $\hat{\mathsf{C}}_{\boldsymbol{x}}(\mathcal{F})$ itself depends on the sample $\boldsymbol{x}$. This result, together with Lemma 1, enables us to use the ERA of the empirical centralization, and Monte-Carlo estimations of it, to derive practical sharp upper-bounds to the SD.

## 3   Uniform convergence bounds

We now introduce novel bounds to the SD using the ERA of the empirical centralization. Before doing so, we must introduce an important technical concept.

**Wimpy variance** The *raw* (i.e., non-centralized) *wimpy variance* $\mathsf{W}^{\mathsf{r}}(\mathcal{F})$ *of $\mathcal{F}$* and the (centralized) *wimpy variance* $\mathsf{W}(\mathcal{F})$ *of $\mathcal{F}$* are key quantities in the study of probabilistic tail bounds to the SD [7, Ch. 11]. They are defined as

$$\mathsf{W}^{\mathsf{r}}(\mathcal{F}) \doteq \sup_{f \in \mathcal{F}} \mathbb{E}_{x \sim \mathcal{D}}\left[(f(x))^2\right], \text{ and } \mathsf{W}(\mathcal{F}) \doteq \sup_{f \in \mathcal{F}} \mathbb{E}_{x \sim \mathcal{D}}\left[(f(x) - \mathbb{E}_{\mathcal{D}}[f])^2\right] \quad . \tag{8}$$

Naturally, the raw wimpy variance is always greater or equal to its centralized counterpart, and potentially much larger. A key identity that we use throughout this work is

$$\mathsf{W}(\mathcal{F}) = \mathsf{W}^{\mathsf{r}}(\mathsf{C}_{\mathcal{D}}(\mathcal{F})) = \mathsf{W}(\mathsf{C}_{\mathcal{D}}(\mathcal{F})) \quad .$$

Empirical estimators on $\boldsymbol{x}$ for the raw wimpy variance and for the wimpy variance are

$$\widehat{\mathsf{W}}^{\mathsf{r}}_{\boldsymbol{x}}(\mathcal{F}) \doteq \sup_{f \in \mathcal{F}} \frac{1}{m} \sum_{i=1}^{m} (f(x_i))^2, \text{ and } \widehat{\mathsf{W}}_{\boldsymbol{x}}(\mathcal{F}) \doteq \sup_{f \in \mathcal{F}} \frac{1}{m} \sum_{i=1}^{m} \left(f(x_i) - \hat{\mathbb{E}}_{\boldsymbol{x}}[f]\right)^2 \quad .$$

To compute the *sample-dependent* bounds to the SD that we introduce later in this section, we develop novel tail bounds to these estimators, which we believe to be of independent interest. Most prior work assumed *known a-priori* bounds to the wimpy variances, but we show that they can be replaced by *empirical* bounds. Maurer and Pontil [18] prove that the sample variance (i.e., when $\mathcal{F}$ is a singleton) is a *weakly self-bounding function* [20]. Our result holds for general $\mathcal{F}$, and is stronger, as we show that the wimpy variance is a *(strongly) self-bounding function* [4, 6] (see also Def. 2 in the supplementary material).

**Theorem 2.** *Suppose $m \geq 2$. Let $\delta \in (0, 1)$. With probability $\geq 1 - \delta$ over the choice of $\boldsymbol{x}$,*

$$\mathsf{W}(\mathcal{F}) \leq \frac{m}{m-1}\widehat{\mathsf{W}}_{\boldsymbol{x}}(\mathcal{F}) + \frac{r^2\ln\frac{1}{\delta}}{m-1} + \sqrt{\left(\frac{r^2\ln\frac{1}{\delta}}{m-1}\right)^2 + \frac{2r^2\frac{m}{m-1}\widehat{\mathsf{W}}_{\boldsymbol{x}}(\mathcal{F})\ln\frac{1}{\delta}}{m-1}} \quad . \tag{9}$$

**Bounds to the SD**  Bousquet [8, Thm. 2.3 (presented here for clarity in a slightly weaker form)] uses the wimpy variance to derive concentration bounds for the SD.

**Theorem 3** (8, Thm. 2.3). *Let $\delta \in (0,1)$. With probability $\geq 1 - \delta$ over the choice of $\boldsymbol{x}$,*

$$\mathsf{SD}(\mathcal{F}, \boldsymbol{x}) \leq \mathbb{E}_{\boldsymbol{x}}\left[\mathsf{SD}(\mathcal{F}, \boldsymbol{x})\right] + \frac{2r \ln \frac{1}{\delta}}{3m} + \sqrt{\frac{2\left(\mathsf{W}(\mathcal{F}) + 4r \, \mathbb{E}_{\boldsymbol{x}}\left[\mathsf{SD}(\mathcal{F}, \boldsymbol{x})\right]\right)\ln \frac{1}{\delta}}{m}} \quad . \tag{10}$$

By plugging the r.h.s. of the symmetrization inequalities (2) and (4) in the r.h.s. of (10), one can obtain bounds that depend on the RA of $\mathcal{F}$ or on the RA of the *distributional centralization* $\mathsf{C}_{\mathcal{D}}(\mathcal{F})$. Neither of these bounds are sample-dependent. Such a bound can be obtained, for example, by using the ERA of $\mathcal{F}$ and a tail bound (e.g., McDiarmid [19]'s inequality or a tail bound for self-bounding functions [6]) on the deviation of the ERA from the RA. The following result states our sample-dependent bound to the SD using the *empirical centralization* $\hat{\mathsf{C}}_{\boldsymbol{x}}(\mathcal{F})$ and tail bounds to the wimpy variance, obtained by combining Lemma 1 and Thms. 1 to 3.

**Theorem 4.** *Assume $m \geq 4$, and let $\eta \in (0,1)$. Take $\nu$ to be the r.h.s. of (9) computed with $\delta = \eta/3$, so $\Pr(\mathsf{W}(\mathcal{F}) > \nu) \leq \eta/3$, and take $\lambda$ to be the r.h.s. of (7) computed with $\delta = \eta/3$, so $\Pr(\mathbb{E}_{\boldsymbol{x}}[\hat{\mathsf{R}}_m(\hat{\mathsf{C}}_{\boldsymbol{x}}(\mathcal{F}), \boldsymbol{x})] > \lambda) \leq \eta/3$. With probability $\geq 1 - \eta$ over the choice of $\boldsymbol{x}$, it holds that*

$$\mathsf{SD}(\mathcal{F}, \boldsymbol{x}) \leq \frac{2\lambda}{1 - 2\mathsf{b}(m)} + \frac{2r \ln \frac{3}{\eta}}{3m} + \sqrt{\frac{2(\nu + 8r\lambda/(1 - 2\mathsf{b}(m)))\ln \frac{3}{\eta}}{m}} \quad .$$

*The r.h.s. is*

$$\frac{2}{1 - 2\mathsf{b}(m)}\hat{\mathsf{R}}_m\left(\hat{\mathsf{C}}_{\boldsymbol{x}}(\mathcal{F}), \boldsymbol{x}\right) + \mathrm{O}\left(\frac{r \ln \frac{1}{\eta}}{m} + \sqrt{\frac{(\mathsf{W}(\mathcal{F}) + r\mathsf{R}_m(\mathsf{C}_{\mathcal{D}}(\mathcal{F}), \mathcal{D}) + r^2/\sqrt{m})\ln \frac{1}{\eta}}{m}}\right) \quad .$$

Is there any advantage in using this bound, i.e., in using empirical centralization, rather than using a bound involving the ERA of $\mathcal{F}$? I.e., how does it compare to the standard bound

$$\mathsf{SD}(\mathcal{F}, \boldsymbol{x}) \leq 2\hat{\mathsf{R}}_m\left(\mathcal{F}, \boldsymbol{x}\right) + \mathrm{O}\left(\frac{q \ln \frac{1}{\eta}}{m} + \sqrt{\frac{(\mathsf{W}(\mathcal{F}) + q\mathsf{R}_m(\mathcal{F}, \mathcal{D}) + q\sqrt{\mathsf{W}^{\mathsf{r}}(\mathcal{F})}/\sqrt{m})\ln \frac{1}{\eta}}{m}}\right) \quad ?$$

We shall see that the $r^2/\sqrt{m}$ and $q\sqrt{\mathsf{W}^{\mathsf{r}}(\mathcal{F})}/\sqrt{m}$ terms are incomparable, though both appear only in transient $\mathrm{O}(m^{-3/4})$ terms, and the remaining differences all favor centralization. Most previous studies on the behavior of SD bounds as functions of the sample size $m$, but we believe that efficient SD bounds for practical applications (e.g., [1, 21, 22, 24, 25]), must improve the dependence also on the other parameters, the *wimpy variance* being the most important. Indeed, developing such bounds is the goal of this work.

First of all, we remark that the dependence on the wimpy variance shown in (10) cannot be improved: any bound to the SD of $\mathcal{F}$ must be $\Omega\sqrt{\mathsf{W}(\mathcal{F})\ln\frac{1}{\delta}/m}$, as can be shown using minimax lower bounds and median-of-means bounds [10, 15]. The question is thus whether the *complexity terms*, i.e., $\hat{\mathsf{R}}_m(\mathcal{F}, \boldsymbol{x})$ and $\hat{\mathsf{R}}_m(\hat{\mathsf{C}}_{\boldsymbol{x}}(\mathcal{F}), \boldsymbol{x})$ can match this lower bound. Lemma 2 answers this question in the *negative* for $\hat{\mathsf{R}}_m(\mathcal{F})$, and in the *positive* for $\hat{\mathsf{R}}_m(\hat{\mathsf{C}}_{\boldsymbol{x}}(\mathcal{F}), \boldsymbol{x})$: the ERA of $\mathcal{F}$ is controlled (in part) by the empirical *raw* wimpy variance, whereas the ERA of $\hat{\mathsf{C}}_{\boldsymbol{x}}(\mathcal{F})$ has corresponding depence on the empirical (centralized) wimpy variance. As with ordinary function variances, the raw wimpy variance can be unboundedly larger than the (centralized) wimpy variance, e.g., in the *constant function family* $\mathcal{F} \doteq \{x \mapsto c\}$.

**Lemma 2.** *For any $\boldsymbol{x} \in \mathcal{X}^m$, it holds*

$$\hat{\mathsf{R}}_m(\mathcal{F}, \boldsymbol{x}) \geq \sqrt{\frac{\widehat{\mathsf{W}^{\mathsf{r}}_{\boldsymbol{x}}(\mathcal{F})}}{2m}} \ \textit{and} \ \hat{\mathsf{R}}_m(\hat{\mathsf{C}}_{\boldsymbol{x}}(\mathcal{F}), \boldsymbol{x}) \geq \sqrt{\frac{\widehat{\mathsf{W}_{\boldsymbol{x}}(\mathcal{F})}}{2m}} \quad .$$

*Furthermore, it holds*

$$\lim_{m \to \infty} \sqrt{m}\mathsf{R}_m(\mathcal{F}, \mathcal{D}) \geq \sqrt{\frac{2}{\pi}\mathsf{W}^{\mathsf{r}}(\mathcal{F})} \ \textit{and} \ \lim_{m \to \infty} \sqrt{m}\mathsf{R}_m(\mathsf{C}_{\mathcal{D}}(\mathcal{F}), \mathcal{D}) \geq \sqrt{\frac{2}{\pi}\mathsf{W}(\mathcal{F})} \quad .$$

To make the result concrete, consider that as soon as $\mathcal{F}$ contains a function $f$ and a "$c$-shifted" version of it $f + c$, for some $c \in \mathbb{R}^+$, then $\sup_{g \in \mathcal{F}} |\hat{\mathbb{E}}_{\boldsymbol{x}}[g]| \geq c/2$, thus $\widehat{\mathsf{W}}_{\boldsymbol{x}}^{\mathsf{r}}(\mathcal{F}) \geq c^2/4$, and from the above lemma, $\hat{\mathsf{R}}_m(\mathcal{F}, \boldsymbol{x}) \geq c/\sqrt{8m}$, but $\hat{\mathsf{R}}_m(\hat{\mathsf{C}}_{\boldsymbol{x}}(\mathcal{F}), \boldsymbol{x})$ does not suffer from this issue.

The significance of Lemma 2 is that a dependence on the (*centralized*) wimpy variance *cannot* be obtained *without* empirical centralization. One must settle for dependence on the *raw* wimpy variance, which can be unboundedly larger than its centralized counterpart. The result also tells us that a dependence on the (centralized) wimpy variance *may* be attained with empirical centralization. We show next that such is indeed the case.

**Optimal dependence on wimpy variance**   The quantity $\hat{\mathsf{R}}_m(\hat{\mathsf{C}}_{\boldsymbol{x}}(\mathcal{F}), \boldsymbol{x})$ is an ERA, thus it can be upper-bounded using Massart's finite-class lemma [16, lemma 5.2]. We now apply this celebrated result to bound the ERA under *empirical centralization* while including the *absolute value* (absent from some presentations) inside the supremum of the ERA.

**Corollary 1.** *Assume that $\mathcal{F}$ is finite. Let $\mathcal{F}_{\pm} \doteq \mathcal{F} \cup \{-f, f \in \mathcal{F}\}$. It holds*

$$\hat{\mathsf{R}}_m(\hat{\mathsf{C}}_{\boldsymbol{x}}(\mathcal{F}), \boldsymbol{x}) \leq \sqrt{\frac{2\widehat{\mathsf{W}}_{\boldsymbol{x}}(\mathcal{F}) \ln |\hat{\mathsf{C}}_{\boldsymbol{x}}(\mathcal{F}_{\pm})|}{m}} \ . \tag{11}$$

The use of $\mathcal{F}_{\pm}$ is needed[2] to handle the absolute value in our definition of the ERA (see (1)). Without empirical centralization, the dependence would be on the raw wimpy variance, which equals the squared $\ell_2$ norm in "classic" presentations of Massart's lemma. Corollary 1 shows that empirical centralization enables *optimal* dependence on the *centralized* wimpy variance, which *cannot be obtained without empirical centralization*, as shown in Lemma 2.

**Monte-Carlo estimation**   The quantity $\hat{\mathsf{R}}_m(\hat{\mathsf{C}}_{\boldsymbol{x}}(\mathcal{F}), \boldsymbol{x})$ is an ERA, so it "suffers" from the usual issue of how to actually compute or bound it in order to bound the SD via Thm. 4. While analytical methods (e.g., Massart's lemma) yield (generally loose) bounds, Monte-Carlo estimation with proper tail bounds gives better results in practice, and it was proposed almost concurrently with the introduction of the ERA [3].

**Definition 1.** *Let $\boldsymbol{\sigma} \in (\pm 1)^{n \times m}$ be a matrix of i.i.d. Rademacher r.v.'s. The* Monte-Carlo *ERA $\hat{\mathsf{R}}_m^n(\mathcal{F}, \boldsymbol{x}, \boldsymbol{\sigma})$ of $\mathcal{F}$ on $\boldsymbol{x}$ w.r.t. $\boldsymbol{\sigma}$ is the quantity*

$$\hat{\mathsf{R}}_m^n(\mathcal{F}, \boldsymbol{x}, \boldsymbol{\sigma}) \doteq \frac{1}{n} \sum_{i=1}^{n} \sup_{f \in \mathcal{F}} \left| \frac{1}{m} \sum_{j=1}^{m} \sigma_{i,j} f(x_i) \right| \ .$$

It clearly holds $\mathbb{E}_{\boldsymbol{\sigma}}[\hat{\mathsf{R}}_m^n(\mathcal{F}, \boldsymbol{x}, \boldsymbol{\sigma})] = \hat{\mathsf{R}}_m(\mathcal{F}, \boldsymbol{x})$. Bartlett and Mendelson [3, Thm. 11] show that the MC-ERA with $n = 1$ is concentrated about the ERA as

$$\Pr_{\boldsymbol{\sigma}} \left( \left| \hat{\mathsf{R}}_m(\mathcal{F}, \boldsymbol{x}) - \hat{\mathsf{R}}_m^1(\mathcal{F}, \boldsymbol{x}, \boldsymbol{\sigma}) \right| \geq \varepsilon \right) \leq 2 \exp \left( \frac{-2m\varepsilon^2}{q^2} \right) \ .$$

The r.h.s. can be used in Thm. 4 inside the definition of $\lambda$ (with the needed adjustment of the confidence parameter $\delta$ using a union bound), thus obtaining an upper bound to the SD using the MC-ERA. The leitmotif of this work is to obtain strong, practical, sample-dependent bounds to the SD, so we derive a novel tail bound to the MC-ERA (Thm. 5) *for general $n$*, where the strong dependence on $q^2$ of the above bound is replaced by a much weaker dependence, primarily on $\mathsf{W}(\mathcal{F})$. This change is similar to how Thm. 3 improves over textbook bounds to the SD that use McDiarmid's bounded difference inequality. Our improved variance-sensitive bound uses a transportation-method inequality due to Samson [26] to *upper bound* the *expectation* of suprema of empirical processes. This result is, to our knowledge, novel, and is worst-case asymptotically equivalent to the McDiarmid bounds, and improves over it when the wimpy variance is small. The bound uses the *empirical maximum $\hat{q}_{\mathcal{F}}(\boldsymbol{x})$ of $\mathcal{F}$ on $\boldsymbol{x}$*, defined as

$$\hat{q}_{\mathcal{F}}(\boldsymbol{x}) \doteq \sup_{f \in \mathcal{F}, x \in \boldsymbol{x}} |f(x)| \qquad (\leq q) \ .$$

**Theorem 5.** *Let $\boldsymbol{\sigma} \in (\pm 1)^{n \times m}$ be a matrix of i.i.d. Rademacher r.v.'s. Let $\delta \in (0, 1)$. With probability at least $1 - \delta$ over the choice of $\boldsymbol{\sigma}$, it holds*

$$\hat{\mathsf{R}}_m(\mathcal{F}, \boldsymbol{x}) \leq \hat{\mathsf{R}}_m^n(\mathcal{F}, \boldsymbol{x}, \boldsymbol{\sigma}) + \frac{2\hat{q}_{\mathcal{F}}(\boldsymbol{x}) \ln \frac{1}{\delta}}{3nm} + \sqrt{\frac{4\widehat{\mathsf{W}}_{\boldsymbol{x}}^{\mathrm{r}}(\mathcal{F}) \ln \frac{1}{\delta}}{nm}} \ . \tag{12}$$

Empirical centralization obtains a dependence on the empirical wimpy variance of $\mathcal{F}$, rather than on the raw (i.e., non-centralized) one. This advantage propagates when using the MC-ERA of the empirical centralization to bound the SD of $\mathcal{F}$. The dependence on the empirical maximum changes from $\hat{q}_{\mathcal{F}}(\boldsymbol{x})$ to $\hat{q}_{\hat{\mathsf{C}}_{\boldsymbol{x}}(\mathcal{F})}(\boldsymbol{x})$, which can be a large improvement (and $\hat{q}_{\hat{\mathsf{C}}_{\boldsymbol{x}}(\mathcal{F})}(\boldsymbol{x}) < 2\hat{q}_{\mathcal{F}}(\boldsymbol{x})$ at most).

**Corollary 2.** *Let $\boldsymbol{\sigma} \in (\pm 1)^{n \times m}$ be a matrix of i.i.d. Rademacher r.v.'s. Let $\delta \in (0, 1)$. With probability at least $1 - \delta$ over the choice of $\boldsymbol{\sigma}$, it holds*

$$\hat{\mathsf{R}}_m(\hat{\mathsf{C}}_{\boldsymbol{x}}(\mathcal{F}), \boldsymbol{x}) \leq \hat{\mathsf{R}}_m^n(\hat{\mathsf{C}}_{\boldsymbol{x}}(\mathcal{F}), \boldsymbol{x}, \boldsymbol{\sigma}) + \frac{2\hat{q}_{\hat{\mathsf{C}}_{\boldsymbol{x}}(\mathcal{F})}(\boldsymbol{x}) \ln \frac{1}{\delta}}{3nm} + \sqrt{\frac{4\widehat{\mathsf{W}}_{\boldsymbol{x}}(\mathcal{F}) \ln \frac{1}{\delta}}{nm}} \ .$$

Although $n = 1$ Monte-Carlo trials are sufficient to match the convergence rate of Thm. 3, the Monte-Carlo estimation error term can still be a significant portion of the total SD bound. For practical usage, particularly with small sample sizes, or when extremely tight bounds are needed, more Monte-Carlo trials (i.e., larger $n$) rapidly reduce the Monte-Carlo estimation error, and this error is soon dominated by the tail bound terms of Thm. 3.

**Example: batch panel of experts** Consider now the *batch panel of experts* problem, where $\mathcal{F}$ is a *finite family* of *experts*, and the task is to select the (approximately) *most accurate* among them, given a sample of *labeled instances*. With the Monte-Carlo method, we may sharply bound the SD whenever evaluating the requisite suprema is computationally feasible, i.e., via *enumeration* of $\mathcal{F}$. Furthermore, we automatically benefit from *data-dependent* and *distribution-dependent* structure, e.g., highly correlated or anticorrelated experts, and *low wimpy variance* over uniformly accurate $\mathcal{F}$. This example immediately extends to *model selection* via *structural risk minimization* if, e.g., the experts are organized into *concentric groups* by some *a priori confidence* or *quality* estimate.

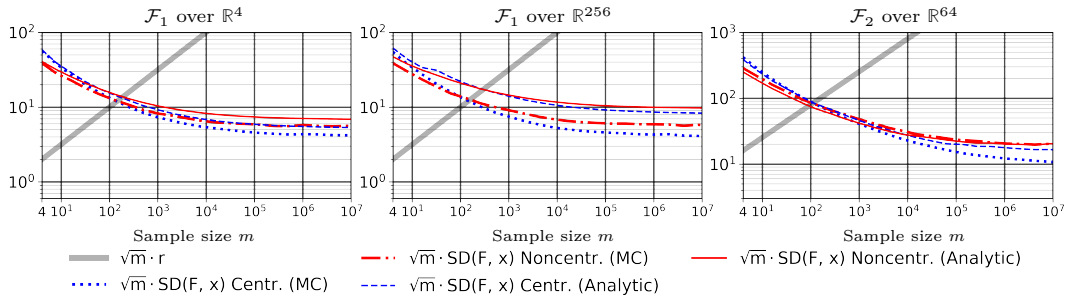

Figure 1: Comparison of SD bounds as functions of the sample size $m$. See the main text for an explanation of the results.

## 4 Experimental evaluation

We performed experiments to evaluate the various bounds presented in the previous sections and compare the bounds to the SD using empirical centralization to those without centralization. The code is included in the supplementary material.

**Function families** We consider the function families $\mathcal{F}_p$, for any $p \geq 1$, containing all unit $\ell_p$-norm-constrained linear functions in $\mathbb{R}^d$, i.e.,

$$\mathcal{F}_p \doteq \{x \mapsto w \cdot x, w \in \mathbb{R}^d \text{ s.t. } \|w\|_p \leq 1\} \ .$$

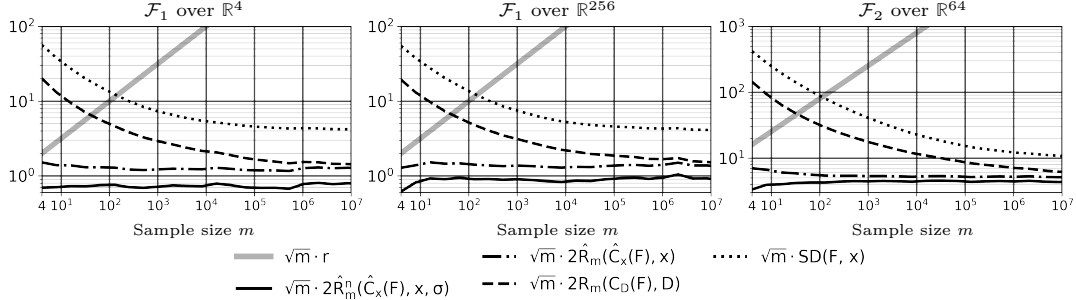

Figure 2: Upper bounds to complexity measures and SD as functions of the sample size $m$. See the main text for details.

These families are of immediate interest in many machine learning settings, such as the analysis of support vector machines and neural networks, as both consist of Lipschitz loss and/or activation functions applied to one or more linear functions (see, e.g., [3] for analysis). Additionally, a bound on the SD of $\mathcal{F}_p$ over distribution $\mathcal{D}$ over $\mathbb{R}^d$ corresponds to the radius of the $\ell_{p/p-1}$ (Hölder dual norm) ball about $\hat{\mathbb{E}}[\boldsymbol{x}]$ in which $\mathbb{E}_{\mathcal{D}}[\boldsymbol{x}] \in \mathbb{R}^d$ falls. Such balls can be used to estimate *covariance matrices*, high-dimensional *sufficient statistics* in graphical models [9], and to learn *equilibria* in simulation-based games [1, 2].

Analytical bounds to the ERA of $\mathcal{F}_p$ on $\boldsymbol{x}$ and Monte-Carlo estimates of it (see Def. 1) are relatively straightforward [27, Lemmas 26.10, 26.11] (see Lemma 5 in the supplementary material). The following lemma extends these results to the empirical centralization.

**Lemma 3.** *Let $\bar{x} \doteq \frac{1}{m}\sum_{i=1}^{m} x_i \in \mathbb{R}^d$. For the $\ell_1$ norm, it holds*

$$\hat{\mathsf{R}}_m(\hat{\mathsf{C}}_{\boldsymbol{x}}(\mathcal{F}_1), \boldsymbol{x}) = \mathbb{E}_{\boldsymbol{\sigma}}\left[\left\|\frac{1}{m}\sum_{i=1}^{m}\boldsymbol{\sigma}_i(x_i - \bar{x})\right\|_\infty\right] \leq \max_i \|x_i - \bar{x}\|_\infty \sqrt{\frac{2\ln(2d)}{m}},$$

*while for the $\ell_2$ norm, it holds*

$$\hat{\mathsf{R}}_m(\hat{\mathsf{C}}_{\boldsymbol{x}}(\mathcal{F}_2), \boldsymbol{x}) = \mathbb{E}_{\boldsymbol{\sigma}}\left[\left\|\frac{1}{m}\sum_{i=1}^{m}\boldsymbol{\sigma}_i(x_i - \bar{x})\right\|_2\right] \leq \max_i \|x_i - \bar{x}\|_2 \frac{1}{\sqrt{m}} \ .$$

Similar bounds are possible for other values of $p$; e.g., by linearity, the case of $p = \infty$ is trivial. Note that in addition to computing MC-ERAs from $\ell_{p/p-1}$ dual norms, we may also compute (raw) empirical wimpy variances from *operator norms* of (raw) *covariance matrices* of $\boldsymbol{x}$. In particular, for $\mathcal{F}_1$, it is easy to show that the wimpy variance is simply the *largest variance* along any *standard basis vector*. Similarly, for $\mathcal{F}_2$, the wimpy variance is simply the *maximum variance* along any *unit vector*, i.e., the *spectral norm* of the covariance matrix.

**Data generation and parameter values** We generated the samples $\boldsymbol{x}$ for our experiments from random distributions over $\mathbb{R}^d$. The ERA of the family $\mathcal{F}_1$ is susceptible to the value of $d$ (see Lemma 3 and Lemma 5 in the supplementary material), so we use $d = 4$ and $d = 256$, while in the case of $\mathcal{F}_2$ the ERA is independent of $d$, so we use $d = 64$. Details of the distributions are in the supplementary material. Range-like quantities (e.g., $q$, $\hat{q}$, $r$) can be computed from the data and/or known a-priori bounds: $r = 1$ for our $\mathcal{F}_1$ experiments and $r = 8$ for the $\mathcal{F}_2$ case. (Raw) wimpy variances correspond to norms of the (raw) covariance matrices used for data generation (see the supplementary material for details). In all experiments, we used $\delta = 0.01$ and $n = 32$ (we comment on this choice below). The sample size $m$ varied from 4 (the minimum possible, due to Lemma 1) to $10^7$.

**A note on results visualization** We present all of our results in plots with *log-log axes*, so that convergence rates are clearly visible as slopes, and constant factors as vertical offsets. The $x$-axis is the sample size $m$. Since we expect asymptotic convergence rates $\propto C/\sqrt{m}$, where $C$ depends on the (possibly raw) wimpy variance of $\mathcal{F}$, $r$, and $\delta$, we plot

all quantities *multiplied by* $\sqrt{m}$. This transformation allows to clearly visualize $\Theta(C/\sqrt{m})$ behaviors as *straight horizontal lines*, and $o(C/\sqrt{m})$ behaviors as (transient) downward slopes. For completeness, we show plots without the scaling by $\sqrt{m}$ in the supplementary material.

**Results** Figure 1 compares four bounds to the SD: using the Monte-Carlo estimate $\hat{\mathsf{R}}_m^n(\hat{\mathsf{C}}_{\boldsymbol{x}}(\mathcal{F}_p), \boldsymbol{x}, \boldsymbol{\sigma})$ for the ERA $\hat{\mathsf{R}}_m(\hat{\mathsf{C}}_{\boldsymbol{x}}(\mathcal{F}_p), \boldsymbol{x})$ of the empirical centralization of $\mathcal{F}_p$ on $\boldsymbol{x}$, using the Monte-Carlo estimate $\hat{\mathsf{R}}_m^n(\mathcal{F}_p, \boldsymbol{x}, \boldsymbol{\sigma})$ for the ERA $\hat{\mathsf{R}}_m(\mathcal{F}_p, \boldsymbol{x})$ of the non-centralized $\mathcal{F}_p$, using analytical bounds to $\hat{\mathsf{R}}_m(\hat{\mathsf{C}}_{\boldsymbol{x}}(\mathcal{F}_p), \boldsymbol{x})$ from Lemma 3, and using analytical bounds to $\hat{\mathsf{R}}_m(\mathcal{F}_p, \boldsymbol{x})$ from Lemma 5 (in the supplementary material). The thicker grey line is the quantity $\sqrt{m}r$; bounds above this line are vacuous.

At very small sample sizes (when all bounds are *vacuous*), the bounds obtained without centralization are sharper than the bounds with empirical centralization, due to the bias-correction of Lemma 1 (see $\xi$ in Thm. 4) and the (fast-decaying) $\Theta(r/m^{3/4})$ term of Thm. 1. Before $m \approx 200$, when bounds become non-vacuous, the advantages of empirical centralization become clear, and increase with the sample size. Recall that each bound is scaled by $\sqrt{m}$, thus all are *asymptotically horizontal*, as $\Theta(C/\sqrt{m})$ terms eventually dominate the bound to the SD, where $C$ varies greatly between bounds and methods. Thus without empirical centralization, obtaining the same bound to the SD would require a larger sample size $m$ than with empirical centralization (this effect can be better observed in the non-$\sqrt{m}$-scaled plots in Fig. 3 in the supplementary material.) The Monte-Carlo estimate, despite using only $n = 32$ Monte-Carlo trials, gives better bounds to the SD than an analytical approach.

In Fig. 2, we drill down on the SD bounds using the Monte-Carlo estimate $\hat{\mathsf{R}}_m^n(\hat{\mathsf{C}}_{\boldsymbol{x}}(\mathcal{F}_p), \boldsymbol{x}, \boldsymbol{\sigma})$ for the ERA $\hat{\mathsf{R}}_m(\hat{\mathsf{C}}_{\boldsymbol{x}}(\mathcal{F}_p), \boldsymbol{x})$ of the empirical centralization of $\mathcal{F}_p$ on $\boldsymbol{x}$, showing this quantity, together with the *upper bounds* to other intermediate quantities, that eventually lead to the SD bound: the ERA $\hat{\mathsf{R}}_m(\hat{\mathsf{C}}_{\boldsymbol{x}}(\mathcal{F}_p), \boldsymbol{x})$ (obtained by applying Thm. 5 to the MC-ERA), the RA $\mathsf{R}_m(\mathcal{F}_p, \mathcal{D})$ (obtained by applying Thm. 1 and Lemma 1 to the bound on the ERA),[3] and SD (obtained by applying the r.h.s. of (4) and Thm. 3 to the bound on the RA).

At small sample sizes, the fast-decaying terms dominate the bounds to the RA and SD, but, true to their nature, quickly become negligible: all bounds are asymptotically $\Theta(C/\sqrt{m})$, where $C$, which in the plots in Fig. 2 appear as the vertical offset of each curve at high sample sizes, depends mostly on the wimpy variance of $\mathcal{F}$ and the range $r$. The bounds that decay as $\Theta\sqrt{\mathsf{W}(\mathcal{F})/m}$ (i.e., the MC-ERA $\to$ ERA and RA $\to$ SD bounds) introduce constant factor terms, manifest as asymptotic vertical gaps, whereas the remaining bounds entirely vanish asymptotically. The gap from the MC-ERA to the ERA would disappear as the number $n$ of Monte-Carlo trials (which we fixed at $n = 32$) increases.

The range and wimpy variances are approximately the same in both $\mathcal{F}_1$ experiments but the MC-ERA are much larger when $d = 256$ because here the RA is essentially the expected largest distance traveled over $d$ random walks, which increases with $d$ (see also Lemma 3).

In conclusion, the results confirm the advantages of empirical centralization to obtain tighter bounds to the SD with optimal dependence on the wimpy variance, while still maintaining the same behavior in terms of the number of samples as bounds not using centralization.

## 5   Conclusions

We develop practical, sharp, sample-dependent probabilistic bounds to the SD through *empirical centralization*, together with novel results on the concentration of the wimpy variance and of Monte-Carlo estimates of the ERA. Our bounds exhibit optimal dependence on the wimpy variance and the same dependence on the number of samples as bounds not using centralization. The results of our experimental evaluation show that the advantage is significant even at small sample sizes, and remains so as the sample size grows. In future work, we will explore the important relationship between centralization and localization [11, 14].

## Statement of broader impact

The goal of our work is to make it possible to get the best possible bounds to the SD as possible from the available data. By reducing the amount of data needed to achieve a certain bound to the SD, we essentially increase the value of each data point, or on the flip side, make each "unit of bound" cheaper. We make essentially no assumption on the family of functions we consider, thus our results are very broadly applicable. As concepts and results from uniform convergence are being used in fields very different than learning (e.g., graph analysis, statistical hypothesis testing, and more, see the Introduction for some references), we believe that enabling machine learning practitioners to *better understand their models*, and scientists in other fields to *make better use of their data* is a positive effort. Certainly, we cannot predict possible misuse of our results, either voluntary or involuntary, in the same way that theoretical results of general applicability are often misused or misapplied (as an example, consider secure cryptographic ciphers that are implemented in the wrong way or used in a cryptographic system is not end-to-end secure).

## Acknowledgments and Disclosure of Funding

This material is based upon work supported by the National Science Foundation under grants 2006765 and RI-1813444, as well as DARPA/AFRL grant FA8750.

## Footnotes

[1] The absolute value can be omitted. All we say can be adapted to this case.

[2]Since $|\hat{\mathsf{C}}_{\boldsymbol{x}}(\mathcal{F}_{\pm})| \leq 2|\hat{\mathsf{C}}_{\boldsymbol{x}}(\mathcal{F})|$, the bound can be reformulated as function of $|\hat{\mathsf{C}}_{\boldsymbol{x}}(\mathcal{F})|$ only.

[3]We do not plot a line for a bound to $\mathbb{E}_{\boldsymbol{x}}[\hat{\mathsf{R}}_m(\hat{\mathsf{C}}_{\boldsymbol{x}}(\mathcal{F}\boldsymbol{x}), \boldsymbol{x})]$ using only Thm. 1 because the fully-multiplicative correction from Lemma 1 is negligible and rapidly decaying.

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
