[Supplementary Material · suppl.pdf]



# A   Proofs

**Lemma 1.** *Suppose $m \geq 4$. Then*

$$\frac{\mathbb{E}_{\boldsymbol{x}}\left[\hat{\mathsf{R}}_m(\hat{\mathsf{C}}_{\boldsymbol{x}}(\mathcal{F}), \boldsymbol{x})\right]}{1 + 2\mathsf{b}(m)} \leq \mathsf{R}_m(\mathsf{C}_{\mathcal{D}}(\mathcal{F}), \mathcal{D}) \leq \frac{\mathbb{E}_{\boldsymbol{x}}\left[\hat{\mathsf{R}}_m(\hat{\mathsf{C}}_{\boldsymbol{x}}(\mathcal{F}), \boldsymbol{x})\right]}{1 - 2\mathsf{b}(m)} \ .$$

*Proof.* We first show the rightmost inequality. Starting from the definition of the RA of the distributional centralization, and then subtracting and adding $\hat{\mathbb{E}}_{\boldsymbol{x}}[f]$, it holds

$$\mathsf{R}_m(\mathsf{C}_{\mathcal{D}}(\mathcal{F}), \mathcal{D}) = \mathbb{E}_{\boldsymbol{\sigma}, \boldsymbol{x}}\left[\sup_{f \in \mathcal{F}}\left|\frac{1}{m}\sum_{i=1}^m \sigma_i\left((f(x_i) - \hat{\mathbb{E}}_{\boldsymbol{x}}[f]) + (\hat{\mathbb{E}}_{\boldsymbol{x}}[f] - \mathbb{E}_{\mathcal{D}}[f])\right)\right|\right] \ .$$

The subadditivity of the supremum and of the absolute value, and the linearity of the expectation allow us to split the r.h.s. into two summands and obtain

$$\mathsf{R}_m(\mathsf{C}_{\mathcal{D}}(\mathcal{F}), \mathcal{D}) \leq \mathbb{E}_{\boldsymbol{\sigma}, \boldsymbol{x}}\left[\sup_{f \in \mathcal{F}}\left|\frac{1}{m}\sum_{i=1}^m \sigma_i(f(x_i) - \hat{\mathbb{E}}_{\boldsymbol{x}}[f])\right|\right] + \mathbb{E}_{\boldsymbol{\sigma}, \boldsymbol{x}}\left[\sup_{f \in \mathcal{F}}\left|\frac{1}{m}\sum_{i=1}^m \sigma_i(\hat{\mathbb{E}}_{\boldsymbol{x}}[f] - \mathbb{E}_{\mathcal{D}}[f])\right|\right] \ .$$

Both terms on the r.h.s. can be seen as expectations w.r.t. $\boldsymbol{x}$ of the ERAs on $\boldsymbol{x}$ of two sample-dependent families: the empirical centralization of $\mathcal{F}$, and the family

$$\mathcal{K}_{\boldsymbol{x}} \doteq \{y \mapsto \hat{\mathbb{E}}_{\boldsymbol{x}}[f] - \mathbb{E}_{\mathcal{D}}[f], f \in \mathcal{F}\} \ .$$

Each function in $\mathcal{K}_{\boldsymbol{x}}$ is *constant*. Thus, we can write

$$\mathsf{R}_m(\mathsf{C}_{\mathcal{D}}(\mathcal{F}), \mathcal{D}) \leq \mathbb{E}_{\boldsymbol{x}}\left[\hat{\mathsf{R}}_m(\hat{\mathsf{C}}_{\boldsymbol{x}}(\mathcal{F}), \boldsymbol{x})\right] + \mathbb{E}_{\boldsymbol{x}}\left[\hat{\mathsf{R}}_m(\mathcal{K}_{\boldsymbol{x}}, \boldsymbol{x})\right] \ . \tag{13}$$

Using (5) and the linearity of expectation we have that, for each $\boldsymbol{x} \in \mathcal{X}^m$, it holds

$$\hat{\mathsf{R}}_m(\mathcal{K}_{\boldsymbol{x}}, \boldsymbol{x}) = \sup_{f \in \mathcal{F}}|\hat{\mathbb{E}}_{\boldsymbol{x}}[f] - \mathbb{E}_{\mathcal{D}}[f]|\mathsf{b}(m) = \mathsf{SD}(\mathcal{F}, \boldsymbol{x})\mathsf{b}(m) = \mathsf{SD}(\mathsf{C}_{\mathcal{D}}(\mathcal{F}), \boldsymbol{x})\mathsf{b}(m), \tag{14}$$

where in the last step we use the fact that the SD is invariant to shifting of functions. Continuing from (13) and using (14) and the rightmost inequality of (4), we obtain

$$\mathsf{R}_m(\mathsf{C}_{\mathcal{D}}(\mathcal{F}), \mathcal{D}) \leq \mathbb{E}_{\boldsymbol{x}}\left[\hat{\mathsf{R}}_m(\hat{\mathsf{C}}_{\boldsymbol{x}}(\mathcal{F}), \boldsymbol{x})\right] + 2\mathsf{R}_m(\mathsf{C}_{\mathcal{D}}(\mathcal{F}), \mathcal{D})\mathsf{b}(m) \ .$$

The hypothesis $m \geq 4$ implies $1 - 2\mathsf{b}(m) > 0$ (see (5)), so we can rewrite the above as

$$\mathsf{R}_m(\mathsf{C}_{\mathcal{D}}(\mathcal{F}), \mathcal{D}) \leq \frac{1}{1 - 2\mathsf{b}(m)}\mathbb{E}_{\boldsymbol{x}}\left[\hat{\mathsf{R}}_m(\hat{\mathsf{C}}_{\boldsymbol{x}}(\mathcal{F}), \boldsymbol{x})\right],$$

which completes the proof of the upper bound.

We next show the lower bound. Starting from the definition of $\hat{\mathsf{R}}_m(\hat{\mathsf{C}}_{\boldsymbol{x}}(\mathcal{F}), \boldsymbol{x})$ and subtracting and adding $\mathbb{E}_{\mathcal{D}}[f]$, it holds

$$\mathbb{E}_{\boldsymbol{x}}[\hat{\mathsf{R}}_m(\hat{\mathsf{C}}_{\boldsymbol{x}}(\mathcal{F}), \boldsymbol{x})] = \mathbb{E}_{\boldsymbol{\sigma}, \boldsymbol{x}}\left[\sup_{f \in \mathcal{F}}\left|\frac{1}{m}\sum_{i=1}^m \sigma_i\left((f(x_i) - \mathbb{E}_{\mathcal{D}}[f]) + (\mathbb{E}_{\mathcal{D}}[f] - \hat{\mathbb{E}}_{\boldsymbol{x}}[f])\right)\right|\right] \ .$$

The subadditivity of the supremum and of the absolute value, and the linearity of the expectation allow us to split the r.h.s. into two summands and obtain

$$\mathbb{E}_{\boldsymbol{x}}[\hat{\mathsf{R}}_m(\hat{\mathsf{C}}_{\boldsymbol{x}}(\mathcal{F}), \boldsymbol{x})] \leq \mathbb{E}_{\boldsymbol{\sigma}, \boldsymbol{x}}\left[\sup_{f \in \mathcal{F}}\left|\frac{1}{m}\sum_{i=1}^m \sigma_i(f(x_i) - \mathbb{E}_{\mathcal{D}}[f])\right|\right]$$

$$+ \mathbb{E}_{\boldsymbol{\sigma}, \boldsymbol{x}}\left[\sup_{f \in \mathcal{F}}\left|\frac{1}{m}\sum_{i=1}^m \sigma_i(\mathbb{E}_{\mathcal{D}}[f] - \hat{\mathbb{E}}_{\boldsymbol{x}}[f])\right|\right] \ . \tag{15}$$

The first term on the r.h.s. is the RA of the *distributional* centralization of $\mathcal{F}$, i.e., it is $\mathsf{R}_m(\mathsf{C}_\mathcal{D}(\mathcal{F}), \mathcal{D})$. The second term is the expectation w.r.t. $\boldsymbol{x}$ of the ERA on $\boldsymbol{x}$ of the family

$$\mathcal{Z}_{\boldsymbol{x}} \doteq \{x \mapsto \mathbb{E}_\mathcal{D}[f] - \hat{\mathbb{E}}_{\boldsymbol{x}}[f], f \in \mathcal{F}\} \ .$$

Each function in $\mathcal{Z}_{\boldsymbol{x}}$ is *constant*. Proceeding in exactly the same way as we did for the family $\mathcal{K}_{\boldsymbol{x}}$ in the proof of the upper bound, we can write

$$\hat{\mathsf{R}}_m(\mathcal{Z}_{\boldsymbol{x}}, \boldsymbol{x}) = \mathsf{SD}(\mathsf{C}_\mathcal{D}(\mathcal{F}), \boldsymbol{x})\mathsf{b}(m) \ . \tag{16}$$

Continuing from (15) and using (16) and the rightmost inequality of (4), we obtain

$$\mathbb{E}_{\boldsymbol{x}}[\hat{\mathsf{R}}_m(\hat{\mathsf{C}}_{\boldsymbol{x}}(\mathcal{F}), \boldsymbol{x})] \leq \mathsf{R}_m(\mathsf{C}_\mathcal{D}(\mathcal{F}), \mathcal{D}) + 2\mathsf{R}_m(\mathsf{C}_\mathcal{D}(\mathcal{F}), \mathcal{D})\mathsf{b}(m) \leq (1 + 2\mathsf{b}(m))\mathsf{R}_m(\mathsf{C}_\mathcal{D}(\mathcal{F}), \mathcal{D}),$$

and our proof is complete. $\qquad\qquad\qquad\qquad\qquad\qquad\qquad\qquad\qquad\qquad\qquad\qquad\qquad\qquad$ $\square$

**Definition 2.** A function $\mathsf{Z} \in \mathcal{X}^m \to \mathbb{R}$ is *($\alpha$,$\beta$)-self-bounding with scale $\gamma$*, for some $\alpha > 0$, $\beta \geq 0$, $\gamma \geq 0$ if for each $j = 1, \ldots, m$, there exists a function $\mathsf{Z}_j \in \mathcal{X}^m \to \mathbb{R}$ such that, for any $\boldsymbol{x} \in \mathcal{X}^m$ it holds that

1. $\mathsf{Z}_j(\boldsymbol{x})$ does not depend on the $j$-th component $x_j$ of $\boldsymbol{x}$; and

2. it holds $\mathsf{Z}_j(\boldsymbol{x}) \leq \mathsf{Z}(\boldsymbol{x}) \leq \mathsf{Z}_j(\boldsymbol{x}) + \gamma$;

Additionally, the functions $\mathsf{Z}_j$, $j = 1, \ldots, m$, must be such that, for any $\boldsymbol{x} \in \mathcal{X}^m$, it holds $\sum_{j=1}^m \left( \mathsf{Z}(\boldsymbol{x}) - \mathsf{Z}_j(\boldsymbol{x}) \right) \leq \alpha\mathsf{Z}(\boldsymbol{x}) + \beta$.

**Theorem 6.** *Let $\mathsf{Z}$ be a function from $\mathcal{X}^m$ to $\mathbb{R}$ that is $(\alpha, \beta)$-self-bounding with scale $\gamma$, for $\alpha \geq 1/3$. Let $\delta \in (0, 1)$ and let $\boldsymbol{x}$ be a collection of $m$ i.i.d. samples from $\mathcal{X}$. With probability at least $1 - \delta$ over the choice of $\boldsymbol{x}$, it holds*

$$\mathbb{E}_{\boldsymbol{x}}[\mathsf{Z}(\boldsymbol{x})] \leq \mathsf{Z}(\boldsymbol{x}) + \alpha\gamma \ln \frac{1}{\delta} + \sqrt{\left( \alpha\gamma \ln \frac{1}{\delta} \right)^2 + 2\gamma(\alpha\mathsf{Z}(\boldsymbol{x}) + \beta) \ln \frac{1}{\delta}} \ . \tag{17}$$

*Additionally, when $\alpha = 1$, we may improve the constants to*

$$\mathbb{E}_{\boldsymbol{x}}[\mathsf{Z}(\boldsymbol{x})] \leq \mathsf{Z}(\boldsymbol{x}) + \frac{2}{3}\gamma \ln \frac{1}{\delta} + \sqrt{\left( \frac{1}{\sqrt{3}}\gamma \ln \frac{1}{\delta} \right)^2 + 2\gamma(\mathsf{Z}(\boldsymbol{x}) + \beta) \ln \frac{1}{\delta}} \ . \tag{18}$$

*Proof.* In both cases, we will assume WLOG $\gamma = 1$. The results then hold by linearity, noting that if $\mathsf{Z}(\cdot)$ is $\alpha$-$\beta$ self-bounding, with scale $\gamma$, then $\frac{1}{\gamma}\mathsf{Z}(\cdot)$ is $\alpha$-$\beta/\gamma$ self-bounding, with scale $1$; the general case thus follows by dividing out $\gamma$, obtaining a bound, and then multiplying through by $\gamma$.

We first show eq. (17). Assume scale $\gamma = 1$. It is known that for $\gamma = 1$, we have for all $\alpha \geq \frac{1}{3}$, as described in [6, Thm. 1], which improves the earlier bounds of [17]

$$\Pr\left( \mathsf{Z}(\boldsymbol{x}) \leq \mathbb{E}_{\boldsymbol{x}}[\mathsf{Z}(\boldsymbol{x})] - \varepsilon \right) \leq \exp\left( \frac{-\varepsilon^2}{2(\alpha\mathbb{E}_{\boldsymbol{x}}[\mathsf{Z}(\boldsymbol{x})] + \beta)} \right) \ . \tag{19}$$

Now, taking $\delta$ equal to the RHS of (19), and solving for $\varepsilon$, this implies that with probability at least $1 - \delta$, we have

$$\mathsf{Z}(\boldsymbol{x}) + \frac{\beta}{\alpha} \geq \mathbb{E}_{\boldsymbol{x}}[\mathsf{Z}(\boldsymbol{x})] + \frac{\beta}{\alpha} - \sqrt{2(\alpha\mathbb{E}_{\boldsymbol{x}}[\mathsf{Z}(\boldsymbol{x})] + \beta) \ln \frac{1}{\delta}} \ .$$

Note that this is a quadratic inequality in $\sqrt{\mathbb{E}_{\boldsymbol{x}}[\mathsf{Z}(\boldsymbol{x})] + \frac{\beta}{\alpha}}$, solving for which (via the quadratic formula) yields nondegenerate solution

$$\mathbb{E}_{\boldsymbol{x}}[\mathsf{Z}(\boldsymbol{x})] \leq \mathsf{Z}(\boldsymbol{x}) + \alpha \ln \frac{1}{\delta} + \sqrt{\left( \alpha \ln \frac{1}{\delta} \right)^2 + 2\alpha(\mathbb{E}_{\boldsymbol{x}}[\mathsf{Z}(\boldsymbol{x})] + \beta) \ln \frac{1}{\delta}} \ .$$

Finally, in the general case, with $\gamma$-scaling, we have

$$\mathbb{E}_{\boldsymbol{x}}\left[\mathsf{Z}(\boldsymbol{x})\right] \leq \mathsf{Z}(\boldsymbol{x}) + \gamma\alpha\ln\frac{1}{\delta} + \sqrt{\left(\gamma\alpha\ln\frac{1}{\delta}\right)^2 + 2\gamma\alpha(\mathbb{E}_{\boldsymbol{x}}\left[\mathsf{Z}(\boldsymbol{x})\right] + \beta)\ln\frac{1}{\delta}} \ .$$

We now show eq. (18) (i.e., assume $\alpha = 1$). Again assume $\gamma = 1$. This result follows via identical logic to the above, this time using the *sub-gamma* form (see Boucheron et al. [7, Ch. 2.1], section 2.1) of the stronger *sub-Poisson* 1-$\beta$ self-bounding function inequality [4, Thm. 1].

In particular, here we have that with probability at least $1 - \delta$,

$$\mathsf{Z}(\boldsymbol{x}) \geq \mathbb{E}_{\boldsymbol{x}}\left[\mathsf{Z}(\boldsymbol{x})\right] + \frac{1}{3}\ln\frac{1}{\delta} - \sqrt{2(\mathbb{E}_{\boldsymbol{x}}\left[\mathsf{Z}(\boldsymbol{x})\right] + \beta)\ln\frac{1}{\delta}} \ ,$$

which by the quadratic formula, yields

$$\mathbb{E}_{\boldsymbol{x}}\left[\mathsf{Z}(\boldsymbol{x})\right] \leq \mathsf{Z}(\boldsymbol{x}) + \frac{2}{3}\ln\frac{1}{\delta} + \sqrt{\left(\frac{\gamma}{\sqrt{3}}\ln\frac{1}{\delta}\right)^2 + 2(\mathbb{E}_{\boldsymbol{x}}\left[\mathsf{Z}(\boldsymbol{x})\right] + \beta)\ln\frac{1}{\delta}} \ .$$

The general result then follows via $\gamma$-scaling.

$\square$

**Theorem 1.** *Suppose $m \geq 1$, and let $\chi \doteq 1 + 2\mathsf{b}(m)$. For any $\delta \in (0, 1)$, with probability at least $1 - \delta$ over the choice of $\boldsymbol{x}$, it holds that*

$$\mathbb{E}_{\boldsymbol{x}}[\hat{\mathsf{R}}_m(\hat{\mathsf{C}}_{\boldsymbol{x}}(\mathcal{F}), \boldsymbol{x})] \leq \hat{\mathsf{R}}_m(\hat{\mathsf{C}}_{\boldsymbol{x}}(\mathcal{F}), \boldsymbol{x}) + \frac{2r\chi\ln\frac{1}{\delta}}{3m} + \sqrt{\left(\frac{r\chi\ln\frac{1}{\delta}}{\sqrt{3}m}\right)^2 + \frac{2r\chi(\hat{\mathsf{R}}_m(\hat{\mathsf{C}}_{\boldsymbol{x}}(\mathcal{F}), \boldsymbol{x}) + r\mathsf{b}(m))\ln\frac{1}{\delta}}{m}} \ . \quad (7)$$

*Proof.* This proof proceeds by showing that $\hat{\mathsf{R}}_m(\hat{\mathsf{C}}_{\boldsymbol{x}}(\mathcal{F}), \boldsymbol{x})$ is a $(1, r\mathsf{b}(m))$-self-bounding function with scale $r\chi/m$, then applying (18) from Thm. 6. First note that the result trivially holds for $m = 1$, as the empirically centralized ERA will always be 0, thus we assume $m \geq 2$ henceforth.

For any $\boldsymbol{x} \in \mathcal{X}^m$, let

$$\mathsf{Y}(\boldsymbol{x}) \doteq \hat{\mathsf{R}}_m(\hat{\mathsf{C}}_{\boldsymbol{x}}(\mathcal{F}), \boldsymbol{x}),$$

and let $\boldsymbol{x}_{\backslash j}$ (resp. $\boldsymbol{\sigma}_{\backslash j}$) denote the $m - 1$-dimensional vector of all but the $j$-th element of $\boldsymbol{x}$ (resp. $\boldsymbol{\sigma}$). Define

$$\mathsf{Y}_j(\boldsymbol{x}) \doteq \frac{m-1}{m}\hat{\mathsf{R}}_{m-1}\left(\hat{\mathsf{C}}_{\boldsymbol{x}_{\backslash j}}(\mathcal{F}), \boldsymbol{x}_{\backslash j}\right) = \mathbb{E}_{\boldsymbol{\sigma}}\left[\sup_{f\in\mathcal{F}}\frac{1}{m}\left|\sum_{i=1, i\neq j}^{m}\sigma_i\left(f(x_i) - \hat{\mathbb{E}}_{\boldsymbol{x}_{\backslash j}}[f]\right)\right|\right] \ .$$

We define these functions for convenience of notation. They will be handy when we later introduce the functions $\mathsf{Z}$ and $\mathsf{Z}_j$, $j = 1, \ldots, m$ that we want to show to be self-bounding.

We now show that $\mathsf{Y}_j(\boldsymbol{x}) \leq \mathsf{Y}(\boldsymbol{x}) + r/m\mathsf{b}(m)$. Starting from the definition of $\mathsf{Y}_j(\boldsymbol{x})$ and adding and subtracting $(f(x_j) - \hat{\mathbb{E}}_{\boldsymbol{x}_{\backslash j}}[f])/2m$ to the argument of the supremum, it holds

$$\mathsf{Y}_j(\boldsymbol{x}) = \mathbb{E}_{\boldsymbol{\sigma}_{\backslash j}}\left[\sup_{f\in\mathcal{F}}\frac{1}{m}\left|\left(\sum_{\substack{i=1\\i\neq j}}^{m}\sigma_i\left(f(x_i) - \hat{\mathbb{E}}_{\boldsymbol{x}_{\backslash j}}[f]\right)\right) + \frac{1}{2}(f(x_j) - \hat{\mathbb{E}}_{\boldsymbol{x}_{\backslash j}}[f]) - \frac{1}{2}(f(x_j) - \hat{\mathbb{E}}_{\boldsymbol{x}_{\backslash j}}[f])\right|\right] \ .$$

Doubling and halving the sum in the argument of the expectation, and leveraging the subadditivity of the supremum and of the absolute value, we obtain

$$\mathsf{Y}_j(\boldsymbol{x}) \leq \mathbb{E}_{\boldsymbol{\sigma}_{\backslash j}}\left[\begin{array}{l}\frac{1}{2}\left(\sup_{f\in\mathcal{F}}\frac{1}{m}\left|\sum_{\substack{i=1\\i\neq j}}^{m}\sigma_i\left(f(x_i) - \hat{\mathbb{E}}_{\boldsymbol{x}_{\backslash j}}[f]\right) + \left(f(x_j) - \hat{\mathbb{E}}_{\boldsymbol{x}_{\backslash j}}[f]\right)\right|\right) \\ + \frac{1}{2}\left(\sup_{f\in\mathcal{F}}\frac{1}{m}\left|\sum_{\substack{i=1\\i\neq j}}^{m}\sigma_i\left(f(x_i) - \hat{\mathbb{E}}_{\boldsymbol{x}_{\backslash j}}[f]\right) - \left(f(x_j) - \hat{\mathbb{E}}_{\boldsymbol{x}_{\backslash j}}[f]\right)\right|\right)\end{array}\right] \ .$$

The two-term sum forming the argument of the outermost expectation is the expectation *w.r.t. only* $\sigma_j$ (i.e., *conditioned* on $\boldsymbol{\sigma}_{\backslash j}$) of the quantity

$$\sup_{f \in \mathcal{F}} \frac{1}{m} \left| \sum_{i=1}^{m} \sigma_i \left( f(x_i) - \hat{\mathbb{E}}_{\boldsymbol{x}_{\backslash j}}[f] \right) \right| \ .$$

Thus, using the law of total expectation, we can write

$$\mathsf{Y}_j(\boldsymbol{x}) \leq \mathbb{E}_{\boldsymbol{\sigma}} \left[ \sup_{f \in \mathcal{F}} \frac{1}{m} \left| \sum_{i=1}^{m} \sigma_i \left( f(x_i) - \hat{\mathbb{E}}_{\boldsymbol{x}_{\backslash j}}[f] \right) \right| \right] \ .$$

By subtracting and adding $\hat{\mathbb{E}}_{\boldsymbol{x}}[f]$ to each term of the sum, and using the subadditivity of the supremum and of the absolute value, and the linearity of the expectation, we obtain

$$\mathsf{Y}_j(\boldsymbol{x}) \leq \underbrace{\mathbb{E}_{\boldsymbol{\sigma}} \left[ \sup_{f \in \mathcal{F}} \frac{1}{m} \left| \sum_{i=1}^{m} \sigma_i \left( f(x_i) - \hat{\mathbb{E}}_{\boldsymbol{x}}[f] \right) \right| \right]}_{=\mathsf{Y}(\boldsymbol{x})} + \mathbb{E}_{\boldsymbol{\sigma}} \left[ \sup_{f \in \mathcal{F}} \frac{1}{m} \left| \sum_{i=1}^{m} \sigma_i \left( \hat{\mathbb{E}}_{\boldsymbol{x}}[f] - \hat{\mathbb{E}}_{\boldsymbol{x}_{\backslash j}}[f] \right) \right| \right] \ .$$

$$(20)$$

The first term on the r.h.s. is $\mathsf{Y}(\boldsymbol{x})$. The second term is the ERA of the sample-dependent family

$$\mathcal{W}_{\boldsymbol{x}} \doteq \left\{ y \mapsto \frac{1}{m}(f(x_j) - \hat{\mathbb{E}}_{\boldsymbol{x}_{\backslash j}}[f]), f \in \mathcal{F} \right\} \ .$$

Each function in $\mathcal{W}_{\boldsymbol{x}}$ is *constant*. Using (5) and the linearity of expectation, like we did in the proof of Lemma 1 for the family $\mathcal{K}_{\boldsymbol{x}}$ (see (14)), it holds

$$\hat{\mathsf{R}}_m(\mathcal{W}_{\boldsymbol{x}}, \boldsymbol{x}) = \frac{1}{m} \sup_{f \in \mathcal{F}} |f(x_j) - \hat{\mathbb{E}}_{\boldsymbol{x}_{\backslash j}}[f]| \mathsf{b}(m) \leq \frac{r}{m}\mathsf{b}(m) \ .$$

Thus, continuing from (20) by incorporating the above fact, it holds

$$\mathsf{Y}_j(\boldsymbol{x}) \leq \mathsf{Y}(\boldsymbol{x}) + \frac{r}{m}\mathsf{b}(m) \ . \tag{21}$$

We now show that $\mathsf{Y}_j(\boldsymbol{x}) \geq \mathsf{Y}(\boldsymbol{x}) - (1 + \mathsf{b}(m))^r/m$. Starting from the definition of $\mathsf{Y}_j$ and adding and removing

$$\frac{1}{m} \left( \sigma_j(f(x_j) - \hat{\mathbb{E}}_{\boldsymbol{x}_{\backslash j}}[f]) \right)$$

to the argument of the supremum, it holds

$$\mathsf{Y}_j(\boldsymbol{x}) = \mathbb{E}_{\boldsymbol{\sigma}} \left[ \sup_{f \in \mathcal{F}} \frac{1}{m} \left| \left( \sum_{\substack{i=1 \\ i \neq j}}^{m} \sigma_i \left( f(x_i) - \hat{\mathbb{E}}_{\boldsymbol{x}_{\backslash j}}[f] \right) \right) + \sigma_j(f(x_j) - \hat{\mathbb{E}}_{\boldsymbol{x}_{\backslash j}}[f]) - \sigma_j(f(x_j) - \hat{\mathbb{E}}_{\boldsymbol{x}_{\backslash j}}[f]) \right| \right] \ .$$

Then, from the triangle inequality and the fact that

$$\sup_{f \in \mathcal{F}} |\sigma_j(f(x_j) - \hat{\mathbb{E}}_{\boldsymbol{x}_{\backslash j}}[f])| \leq r,$$

we obtain

$$\mathsf{Y}_j(\boldsymbol{x}) \geq \mathbb{E}_{\boldsymbol{\sigma}} \left[ \sup_{f \in \mathcal{F}} \frac{1}{m} \left| \sum_{i=1}^{m} \sigma_i \left( f(x_i) - \hat{\mathbb{E}}_{\boldsymbol{x}_{\backslash j}}[f] \right) \right| \right] - \frac{r}{m} \ .$$

From here, we add and subtract $\sigma_i \hat{\mathbb{E}}_{\boldsymbol{x}}[f]$ to each term of the sum, and then use the triangle inequality, the subadditivity of the supremum, and the linearity of expectation, to obtain

$$\mathsf{Y}_j(\boldsymbol{x}) \geq \underbrace{\mathbb{E}_{\boldsymbol{\sigma}} \left[ \sup_{f \in \mathcal{F}} \frac{1}{m} \left| \sum_{i=1}^{m} \sigma_i \left( f(x_i) - \hat{\mathbb{E}}_{\boldsymbol{x}}[f] \right) \right| \right]}_{=\mathsf{Y}(\boldsymbol{x})} - \mathbb{E}_{\boldsymbol{\sigma}} \left[ \sup_{f \in \mathcal{F}} \frac{1}{m} \left| \sum_{i=1}^{m} \sigma_i \left( \hat{\mathbb{E}}_{\boldsymbol{x}}[f] - \hat{\mathbb{E}}_{\boldsymbol{x}_{\backslash j}}[f] \right) \right| \right] - \frac{r}{m} \ .$$

The second term on the r.h.s. is again the ERA of a family of constant functions, each of them taking value at most $r/m$. Thus using (5), it follows that

$$\mathsf{Y}_j(\boldsymbol{x}) \geq \mathsf{Y}(\boldsymbol{x}) - (1 + \mathsf{b}(m))\frac{r}{m} \ .$$

Combining the above and (21), we obtain

$$\mathsf{Y}(\boldsymbol{x}) - (1 + \mathsf{b}(m))\frac{r}{m} \leq \mathsf{Y}_j(\boldsymbol{x}) \leq \mathsf{Y}(\boldsymbol{x}) + \frac{r}{m}\mathsf{b}(m) \ . \tag{22}$$

We now show that

$$\sum_{j=1}^{m} \left( \mathsf{Y}(\boldsymbol{x}) - \mathsf{Y}_j(\boldsymbol{x}) \right) \leq \mathsf{Y}(\boldsymbol{x}) \ . \tag{23}$$

Starting from the definition of the $\mathsf{Y}_j$ functions, and using the linearity of expectation and the subadditivity of the supremum

$$\sum_{j=1}^{m} \mathsf{Y}_j(\boldsymbol{x}) = \sum_{j=1}^{m} \mathbb{E}_{\boldsymbol{\sigma}} \left[ \sup_{f \in \mathcal{F}} \frac{1}{m} \left| \sum_{i=1, i \neq j}^{m} \sigma_i \left( f(x_i) - \hat{\mathbb{E}}_{\boldsymbol{x}_{\backslash j}}[f] \right) \right| \right]$$

$$\geq \mathbb{E}_{\boldsymbol{\sigma}} \left[ \sup_{f \in \mathcal{F}} \frac{1}{m} \left| \sum_{j=1}^{m} \sum_{i=1, i \neq j}^{m} \sigma_i \left( f(x_i) - \hat{\mathbb{E}}_{\boldsymbol{x}_{\backslash j}}[f] \right) \cdot \right| \right] \ .$$

We rearrange the terms in the double sums, and use the linearity of expectation to obtain

$$\sum_{j=1}^{m} \mathsf{Y}_j(\boldsymbol{x}) \geq \mathbb{E}_{\boldsymbol{\sigma}} \left[ \sup_{f \in \mathcal{F}} \frac{1}{m} \left| (m-1) \sum_{i=1}^{m} \sigma_i \left( f(x_i) - \hat{\mathbb{E}}_{\boldsymbol{x}}[f] \right) \right| \right]$$

$$\geq (m-1) \mathbb{E}_{\boldsymbol{\sigma}} \left[ \sup_{f \in \mathcal{F}} \frac{1}{m} \left| \sum_{i=1}^{m} \sigma_i \left( f(x_i) - \hat{\mathbb{E}}_{\boldsymbol{x}}[f] \right) \right| \right] ,$$

which completes our proof of (23), as the last expectation is $\mathsf{Y}(\boldsymbol{x})$.

Define now the functions

$$\mathsf{Z}(\boldsymbol{x}) \doteq \mathsf{Y}(\boldsymbol{x}) \text{ and } \mathsf{Z}_j(\boldsymbol{x}) \doteq \mathsf{Y}_j(\boldsymbol{x}) - \frac{r}{m}\mathsf{b}(m) \text{ for each } j = 1, \dots, m \ .$$

The value of $\mathsf{Z}_j(\boldsymbol{x})$ clearly does not dependent on the $j$-th component of $\boldsymbol{x}$. Also, from (22) it follows that

$$\mathsf{Z}_j(\boldsymbol{x}) \leq \mathsf{Z}(\boldsymbol{x}) \leq \mathsf{Z}_j(\boldsymbol{x}) + (1 + 2\mathsf{b}(m))\frac{r}{m} \text{ for each } j = 1, \dots, m \ .$$

A consequence of (23) is finally that

$$\sum_{j=1}^{m} \left( \mathsf{Z}(\boldsymbol{x}) - \mathsf{Z}_j(\boldsymbol{x}) \right) \leq \mathsf{Z}(\boldsymbol{x}) + r\mathsf{b}(m) \ .$$

Thus $\mathsf{Z}$, i.e., $\hat{\mathsf{R}}_m(\hat{\mathsf{C}}_{\boldsymbol{x}}(\mathcal{F}), \boldsymbol{x})$, is a $(1, r\mathsf{b}(m))$-self-bounding function with scale $(1 + 2\mathsf{b}(m))r/m$. An application of (18) from Thm. 6 completes the proof. □

Before proving Thm. 2, we need the following lemma.

**Lemma 4.** *It holds*

$$\mathsf{W}(\mathcal{F}) \leq \frac{m}{m-1} \mathbb{E}_{\boldsymbol{x}} [\hat{\mathsf{W}}_{\boldsymbol{x}}(\mathcal{F})] \ .$$

*Proof.* Using Bessel's correction, we can rewrite the definition of wimpy variance to use the empirical expectation as

$$\mathsf{W}(\mathcal{F}) = \sup_{f \in \mathcal{F}} \mathbb{E}_{\boldsymbol{x}} \left[ \frac{1}{m} \sum_{i=1}^{m} \left( f(x_i) - \mathbb{E}_{\mathcal{D}}[f] \right)^2 \right] = \sup_{f \in \mathcal{F}} \mathbb{E}_{\boldsymbol{x}} \left[ \frac{1}{m-1} \sum_{i=1}^{m} \left( f(x_i) - \hat{\mathbb{E}}_{\boldsymbol{x}}[f] \right)^2 \right] \ .$$

An application of Jensen's inequality gives

$$\mathsf{W}(\mathcal{F}) \leq \mathbb{E}_{\boldsymbol{x}} \left[ \underbrace{\sup_{f \in \mathcal{F}} \frac{1}{m-1} \sum_{i=1}^{m} \left( f(x_i) - \hat{\mathbb{E}}_{\boldsymbol{x}}[f] \right)^2}_{= \frac{m}{m-1} \hat{\mathsf{W}}_{\boldsymbol{x}}(\mathcal{F})} \right] \ .$$

□

**Theorem 2.** *Suppose $m \geq 2$. Let $\delta \in (0,1)$. With probability $\geq 1 - \delta$ over the choice of $\boldsymbol{x}$,*

$$\mathsf{W}(\mathcal{F}) \leq \frac{m}{m-1}\widehat{\mathsf{W}}_{\boldsymbol{x}}(\mathcal{F}) + \frac{r^2 \ln\frac{1}{\delta}}{m-1} + \sqrt{\left(\frac{r^2 \ln\frac{1}{\delta}}{m-1}\right)^2 + \frac{2r^2 \frac{m}{m-1}\widehat{\mathsf{W}}_{\boldsymbol{x}}(\mathcal{F}) \ln\frac{1}{\delta}}{m-1}} \quad . \tag{9}$$

*Proof.* This proof proceeds by showing that $\widehat{\mathsf{W}}_{\boldsymbol{x}}(\mathcal{F})$ is a $(m/m-1, 0)$-self-bounding with scale $r^2/m$, then applying Lemma 4, and finally (17) from Thm. 6.

Let $\boldsymbol{x}_{\setminus j}$ denote the vector $\boldsymbol{x}$ with the $j$-th component removed, as we defined it also in the proof for Thm. 1. Let $\hat{\mathsf{V}}_{\boldsymbol{x}}[f]$ denote the (unbiased) sample variance of $f$ over $\boldsymbol{x}$, i.e.,

$$\hat{\mathsf{V}}_{\boldsymbol{x}}[f] \doteq \frac{1}{m-1}\sum_{i=1}^{m}\left(f(x_i) - \hat{\mathbb{E}}_{\boldsymbol{x}}[f]\right)^2 \quad .$$

Define

$$\mathsf{Z}(\boldsymbol{x}) \doteq \frac{m}{m-1}\widehat{\mathsf{W}}_{\boldsymbol{x}}(\mathcal{F}) = \sup_{f \in \mathcal{F}}\hat{\mathsf{V}}_{\boldsymbol{x}}[f] = \sup_{f \in \mathcal{F}}\frac{1}{m-1}\sum_{i=1}^{m}\left(f(x_i) - \hat{\mathbb{E}}_{\boldsymbol{x}}[f]\right)^2$$

and

$$\mathsf{Z}_j(\boldsymbol{x}) \doteq \sup_{f \in \mathcal{F}}\frac{1}{m-1}\sum_{i=1, i\neq j}^{m}\left(f(x_i) - \hat{\mathbb{E}}_{\boldsymbol{x}_{\setminus j}}[f]\right)^2 \quad . \tag{24}$$

We first show that

$$\mathsf{Z}_j(\boldsymbol{x}) = \sup_{f \in \mathcal{F}}\left[\hat{\mathsf{V}}_{\boldsymbol{x}}[f] - \frac{1}{m}\left(f(x_j) - \hat{\mathbb{E}}_{\boldsymbol{x}_{\setminus j}}[f]\right)^2\right], \tag{25}$$

as this form comes in handy many times. Starting from the definition of $\mathsf{Z}_j$ in (24), we add and subtract $\frac{1}{m-1}(f(x_j) - \hat{\mathbb{E}}_{\boldsymbol{x}_{\setminus j}}[f])^2$ to the argument of the supremum, and then add and subtract $\hat{\mathbb{E}}_{\boldsymbol{x}}[f]$ to the argument of the sum, to obtain:

$$\mathsf{Z}_j(\boldsymbol{x}) = \sup_{f \in \mathcal{F}}\frac{1}{m-1}\left[\left(\sum_{i=1}^{m}(f(x_i) - \hat{\mathbb{E}}_{\boldsymbol{x}_{\setminus j}}[f])^2\right) - (f(x_j) - \hat{\mathbb{E}}_{\boldsymbol{x}_{\setminus j}}[f])^2\right]$$

$$= \sup_{f \in \mathcal{F}}\frac{1}{m-1}\left[\left(\sum_{i=1}^{m}\left((f(x_i) - \hat{\mathbb{E}}_{\boldsymbol{x}}[f]) + (\hat{\mathbb{E}}_{\boldsymbol{x}}[f] - \hat{\mathbb{E}}_{\boldsymbol{x}_{\setminus j}}[f])\right)^2\right) - (f(x_j) - \hat{\mathbb{E}}_{\boldsymbol{x}_{\setminus j}}[f])^2\right]$$

By expressing the square in the argument of the sum, separating the three resulting terms in three distinct sums (associative property of the sum), and noticing that one of these sum is $\sum_{i=1}^{m}(f(x_i) - \hat{\mathbb{E}}_{\boldsymbol{x}}[f]) = 0$, and another has argument $(\hat{\mathbb{E}}_{\boldsymbol{x}}[f] - \hat{\mathbb{E}}_{\boldsymbol{x}_{\setminus j}}[f])^2$ independent from $i$, we obtain

$$\mathsf{Z}_j(\boldsymbol{x}) = \sup_{f \in \mathcal{F}}\frac{1}{m-1}\left[\underbrace{\left(\sum_{i=1}^{m}(f(x_i) - \hat{\mathbb{E}}_{\boldsymbol{x}}[f])^2\right)}_{=(m-1)\hat{\mathsf{V}}_{\boldsymbol{x}}[f]} + m(\hat{\mathbb{E}}_{\boldsymbol{x}}[f] - \hat{\mathbb{E}}_{\boldsymbol{x}_{\setminus j}}[f])^2 - (f(x_j) - \hat{\mathbb{E}}_{\boldsymbol{x}_{\setminus j}}[f])^2\right] \quad .$$

It holds $\hat{\mathbb{E}}_{\boldsymbol{x}}[f] = \frac{1}{m}f(x_j) + \frac{m-1}{m}\hat{\mathbb{E}}_{\boldsymbol{x}_{\setminus j}}[f]$, so we have

$$\mathsf{Z}_j(\boldsymbol{x}) = \sup_{f \in \mathcal{F}}\frac{1}{m-1}\left[(m-1)\hat{\mathsf{V}}_{\boldsymbol{x}}[f] + m\left(\frac{1}{m}f(x_j) - \frac{1}{m}\hat{\mathbb{E}}_{\boldsymbol{x}_{\setminus j}}[f]\right)^2 - (f(x_j) - \hat{\mathbb{E}}_{\boldsymbol{x}_{\setminus j}}[f])^2\right] \quad .$$

The identity in (25) then follows through simple algebraic steps.

We want to show that $\mathsf{Z}$ is a $(m/m-1, 0)$-self-bounding function with scale $r^2/m$ (see Def. 2). By definition of $\mathsf{Z}_j$ in (24), the value of $\mathsf{Z}_j(\boldsymbol{x})$ does not depend on the $j$-th component of $\boldsymbol{x}$, as required by the first point in Def. 2.

We now show that, for any $j = 1, \ldots, m$, it holds,

$$\mathsf{Z}_j(\boldsymbol{x}) \leq \mathsf{Z}(\boldsymbol{x}) \leq \mathsf{Z}_j(\boldsymbol{x}) + \frac{r^2}{m} \text{ for any } \boldsymbol{x} \in \mathcal{X}^m, \tag{26}$$

as required by the second point in Def. 2. The leftmost inequality follows from the definitions of $\mathsf{Z}$ and $\mathsf{Z}_j$. To show the rightmost inequality, we start from (25), and use the subadditivity of the supremum to obtain

$$\mathsf{Z}_j(\boldsymbol{x}) \geq \underbrace{\left[\left(\sup_{f \in \mathcal{F}} \hat{\mathsf{V}}_{\boldsymbol{x}}[f]\right)}_{=\mathsf{Z}(\boldsymbol{x})} - \left(\sup_{f \in \mathcal{F}} \frac{1}{m}\left(f(x_j) - \hat{\mathbb{E}}_{\boldsymbol{x}_{\backslash j}}[f]\right)^2\right)\right] \ .$$

The rightmost supremum is always smaller than $r^2/m$ because $|f(x_j) - \hat{\mathbb{E}}_{\boldsymbol{x}_{\backslash j}}[f]| \leq r$, thus we have obtained the rightmost inequality in (26).

We now show that, for any $\boldsymbol{x} \in \mathcal{X}^m$, it holds

$$\sum_{i=1}^{m} \left(\mathsf{Z}(\boldsymbol{x}) - \mathsf{Z}_j(\boldsymbol{x})\right) \leq \frac{m}{m-1} \mathsf{Z}(\boldsymbol{x}),$$

as in the last requirement of Def. 2. Starting again from (25) and using the subadditivity of the supremum, it holds

$$\sum_{j=1}^{m} \mathsf{Z}_j(\boldsymbol{x}) = \sum_{j=1}^{m} \sup_{f \in \mathcal{F}} \left[\hat{\mathsf{V}}_{\boldsymbol{x}}[f] - \frac{1}{m}\left(f(x_j) - \hat{\mathbb{E}}_{\boldsymbol{x}_{\backslash j}}[f]\right)^2\right] \geq \sup_{f \in \mathcal{F}} \sum_{j=1}^{m} \left[\hat{\mathsf{V}}_{\boldsymbol{x}}[f] - \frac{1}{m}\left(f(x_j) - \hat{\mathbb{E}}_{\boldsymbol{x}_{\backslash j}}[f]\right)^2\right] \ .$$

By simple algebra we then get

$$\sum_{j=1}^{m} \mathsf{Z}_j(\boldsymbol{x}) \geq \sup_{f \in \mathcal{F}} \left[m\hat{\mathsf{V}}_{\boldsymbol{x}}[f] - \frac{1}{m}\sum_{j=1}^{m}\left(f(x_j) - \hat{\mathbb{E}}_{\boldsymbol{x}_{\backslash j}}[f]\right)^2\right] \ .$$

From here, we use the fact that

$$\hat{\mathbb{E}}_{\boldsymbol{x}_{\backslash j}}[f] = \frac{1}{m-1}\left(m\hat{\mathbb{E}}_{\boldsymbol{x}}[f] - f(x_j)\right),$$

to get

$$\sum_{j=1}^{m} \mathsf{Z}_j(\boldsymbol{x}) \geq \sup_{f \in \mathcal{F}} \left[m\hat{\mathsf{V}}_{\boldsymbol{x}}[f] - \frac{1}{m}\sum_{j=1}^{m}\left(\frac{m}{m-1}f(x_j) - \frac{m}{m-1}\hat{\mathbb{E}}_{\boldsymbol{x}}[f]\right)^2\right] \ .$$

Now by simplifying some terms on the r.h.s., we obtain

$$\sum_{j=1}^{m} \mathsf{Z}_j(\boldsymbol{x}) \geq \sup_{f \in \mathcal{F}} \left[m\hat{\mathsf{V}}_{\boldsymbol{x}}[f] - \frac{m}{(m-1)}\underbrace{\frac{1}{m-1}\sum_{j=1}^{m}\left(f(x_j) - \hat{\mathbb{E}}_{\boldsymbol{x}}[f]\right)^2}_{=\hat{\mathsf{V}}_{\boldsymbol{x}}[f]}\right] \ .$$

Collecting terms and using the original definition of $\mathsf{Z}$ results in

$$\sum_{j=1}^{m} \mathsf{Z}_j(\boldsymbol{x}) \geq \left(m - \frac{m}{m-1}\right)\mathsf{Z}(\boldsymbol{x}) \ .$$

Thus,

$$\sum_{j=1}^{m} \left(\mathsf{Z}(\boldsymbol{x}) - \mathsf{Z}_j(\boldsymbol{x})\right) \leq m\mathsf{Z}(\boldsymbol{x}) - \left(m - \frac{m}{m-1}\right)\mathsf{Z}(\boldsymbol{x}) \leq \frac{m}{m-1}\mathsf{Z}(\boldsymbol{x}),$$

which concludes our proof that $\mathsf{Z}$, is $(m/m-1, 0)$-self-bounding with scale $r^2/m$.

We now use the above fact to prove the thesis. A consequence of Lemma 4 is

$$\Pr_{\boldsymbol{x}}\left(\widehat{\mathsf{W}}_{\boldsymbol{x}}(\mathcal{F}) \leq \mathsf{W}(\mathcal{F}) - \varepsilon\right) \leq \Pr_{\boldsymbol{x}}\left(\widehat{\mathsf{W}}_{\boldsymbol{x}}(\mathcal{F}) \leq \frac{m}{m-1}\mathbb{E}_{\boldsymbol{x}}[\widehat{\mathsf{W}}_{\boldsymbol{x}}(\mathcal{F})] - \varepsilon\right) \ .$$

From here, we use the definition

$$\mathsf{Z}(\boldsymbol{x}) = \frac{m}{m-1}\widehat{\mathsf{W}}_{\boldsymbol{x}}(\mathcal{F})$$

and apply (17) from Thm. 6 to obtain the thesis. $\qquad\square$

The constants in this bound are somewhat sub-optimal, as there is a significant gap between the best-known (sub-Poisson) tails for $(1,0)$-self-bounding and the best-known (sub-gamma) tails for $(1+\varepsilon, 0)$-self-bounding functions. We hope that future work leads to refined analysis of tail bounds for $(\alpha, 0)$-self-bounding functions that decay gracefully as $\alpha$ exceeds 1.

**Lemma 2.** *For any $\boldsymbol{x} \in \mathcal{X}^m$, it holds*

$$\hat{\mathsf{R}}_m(\mathcal{F}, \boldsymbol{x}) \geq \sqrt{\frac{\widehat{\mathsf{W}}_{\boldsymbol{x}}^{\mathsf{r}}(\mathcal{F})}{2m}} \ \text{ and } \ \hat{\mathsf{R}}_m(\hat{\mathsf{C}}_{\boldsymbol{x}}(\mathcal{F}), \boldsymbol{x}) \geq \sqrt{\frac{\widehat{\mathsf{W}}_{\boldsymbol{x}}(\mathcal{F})}{2m}} \ .$$

*Furthermore, it holds*

$$\lim_{m \to \infty} \sqrt{m} \mathsf{R}_m(\mathcal{F}, \mathcal{D}) \geq \sqrt{\frac{2}{\pi} \mathsf{W}^{\mathsf{r}}(\mathcal{F})} \ \text{ and } \ \lim_{m \to \infty} \sqrt{m} \mathsf{R}_m(\mathsf{C}_{\mathcal{D}}(\mathcal{F}), \mathcal{D}) \geq \sqrt{\frac{2}{\pi} \mathsf{W}(\mathcal{F})} \ .$$

*Proof.* From the subadditivity of the supremum, it holds that

$$\hat{\mathsf{R}}_m(\mathcal{F}, \boldsymbol{x}) \geq \sup_{f \in \mathcal{F}} \mathbb{E}_{\boldsymbol{\sigma}} \left[ \left| \frac{1}{m} \sum_{i=1}^m \sigma_i f(x_i) \right| \right] \ .$$

An application of Khintchine's inequality [12] gives

$$\hat{\mathsf{R}}_m(\mathcal{F}, \boldsymbol{x}) \geq \sup_{f \in \mathcal{F}} \frac{1}{\sqrt{2}} \sqrt{\frac{\|f(\boldsymbol{x})\|_2^2}{m^2}},$$

where $f(\boldsymbol{x})$ denotes the $m$-dimensional vector of values of $f$ on $\boldsymbol{x}$. The proof of the leftmost inequality in the thesis ends by noting that

$$\widehat{\mathsf{W}}_{\boldsymbol{x}}^{\mathsf{r}}(\mathcal{F}) = \frac{\|f(\boldsymbol{x})\|_2^2}{m} \ .$$

The rightmost inequality is then a corollary, using the identity $\widehat{\mathsf{W}}_{\boldsymbol{x}}^{\mathsf{r}}(\hat{\mathsf{C}}_{\boldsymbol{x}}(\mathcal{F})) = \widehat{\mathsf{W}}_{\boldsymbol{x}}(\mathcal{F})$.

The asymptotic lower bounds follow by replacing the Khintchine's inequality step with an application of the central limit theorem. □

Before proving Thm. 5 we need to introduce an important technical result. For any $u \in \mathbb{R}$, let $\mathsf{h}(u) \doteq (1+u) \ln(1+u) - u$, and let $(u)_+ \doteq \max(0, u)$.

**Theorem 7** (Samson's bound, [7, Thm. 12.11]). *Let $\mathcal{Q}_1, \dots, \mathcal{Q}_m$ be possibly different probability distributions over a domain $\mathcal{Y}$. Let $\mathcal{G} \subseteq \mathcal{X} \to [-1, 1]$. Furthermore, assume that for each $g \in \mathcal{G}$ and $i \in \{1, \dots, m\}$, it holds $\mathbb{E}_{\mathcal{Q}_i}[g] = 0$. Now, for any $\boldsymbol{y} \in \mathcal{Y}^m$, let*

$$\mathsf{Z}(\boldsymbol{y}) \doteq \sup_{g \in \mathcal{G}} \sum_{i=1}^m g(y_i) \ \text{ and } \ S^2 \doteq \mathbb{E}_{\boldsymbol{y}} \left[ \sup_{g \in \mathcal{F}} \sum_{i=1}^m \mathbb{E}_{y_i' \sim \mathcal{Q}_i} \left[ \left( (g(y_i) - g(y_i'))_+ \right)^2 \right] \right] \ .$$

*Let $\boldsymbol{y} \in \mathcal{Y}^m$, with each $y_i \sim \mathcal{Q}_i$, independently (but not necessarily identically, since the distributions may be different). It holds*[4]

$$\Pr_{\boldsymbol{y}} \left( \mathsf{Z}(\boldsymbol{y}) \leq \mathbb{E}_{\mathcal{Q}_{1:m}}[\mathsf{Z}] - \varepsilon \right) \leq \exp \left( -\frac{S^2}{4} \mathsf{h} \left( \frac{2\varepsilon}{S^2} \right) \right) \ . \tag{27}$$

**Theorem 5.** *Let $\boldsymbol{\sigma} \in (\pm 1)^{n \times m}$ be a matrix of i.i.d. Rademacher r.v.'s. Let $\delta \in (0, 1)$. With probability at least $1 - \delta$ over the choice of $\boldsymbol{\sigma}$, it holds*

$$\hat{\mathsf{R}}_m(\mathcal{F}, \boldsymbol{x}) \leq \hat{\mathsf{R}}_m^n(\mathcal{F}, \boldsymbol{x}, \boldsymbol{\sigma}) + \frac{2\hat{q}_{\mathcal{F}}(\boldsymbol{x}) \ln \frac{1}{\delta}}{3nm} + \sqrt{\frac{4 \widehat{\mathsf{W}}_{\boldsymbol{x}}^{\mathsf{r}}(\mathcal{F}) \ln \frac{1}{\delta}}{nm}} \ . \tag{12}$$

*Proof.* Without loss of generality, we assume that $\hat{q}_{\mathcal{F}}(\boldsymbol{x}) = 1$. The general case then follows via scaling.

Let

$$\mathsf{Z}(\boldsymbol{\sigma}) \doteq nm\widehat{\mathsf{R}}_m^n(\mathcal{F}, \boldsymbol{x}, \boldsymbol{\sigma}) = \sum_{j=1}^{n} \sup_{f \in \mathcal{F}} \left| \sum_{i=1}^{m} \sigma_{j,i} f(x_i) \right| \ .$$

It holds $\mathbb{E}_{\boldsymbol{\sigma}}[\mathsf{Z}] = nm\widehat{\mathsf{R}}_m(\mathcal{F}, \boldsymbol{x})$.

We first show that we can apply Samson's bound (Thm. 7) to $\mathsf{Z}$, i.e., to the scaled MC-ERA. Consider the function family $\mathcal{F}_{\pm}$ introduced in Coro. 1, and consider the $n$-times Cartesian product of $\mathcal{F}_{\pm}$ with itself

$$(\mathcal{F}_{\pm})^n = \underbrace{\mathcal{F}_{\pm} \times \cdots \times \mathcal{F}_{\pm}}_{n \text{ times}} \ .$$

We use $\boldsymbol{f} = (f_1, \ldots, f_n)$ to denote an element of $(\mathcal{F}_{\pm})^n$. Now, define the family

$$\mathcal{G} \doteq \{g(\sigma_{j,i}) \doteq \sigma_{j,i} f_j(x_i), \boldsymbol{f} \in (\mathcal{F}_{\pm})^n\} \ .$$

The functions in $\mathcal{G}$ have domain $\mathcal{Y} = \{-1, 1\}$ and values in $[-1, 1]$. It holds

$$\mathsf{Z}(\boldsymbol{\sigma}) = \sup_{\boldsymbol{f} \in (\mathcal{F}_{\pm})^n} \sum_{j=1}^{n} \sum_{i=1}^{m} \sigma_{j,i} f_j(x_i) = \sup_{g \in \mathcal{G}} \sum_{(j,i) \in \{1, \ldots n\} \times \{1, \ldots, m\}} g(\sigma_{j,i}) \ . \tag{28}$$

Thus $\mathsf{Z}$ has the form required by Thm. 7.

Let $\boldsymbol{\sigma}'$ denote a second $n \times m$ i.i.d. Rademacher matrix (like $\boldsymbol{\sigma}$), and define

$$S^2 \doteq \mathbb{E}_{\boldsymbol{\sigma}} \left[ \sup_{\boldsymbol{f} \in (\mathcal{F}_{\pm})^n} \sum_{j=1}^{n} \sum_{i=1}^{m} \mathbb{E}_{\sigma'_{j,i}} \left[ \left( (\sigma_{j,i} f_j(x_i) - \sigma'_{j,i} f_j(x_i))_+ \right)^2 \right] \right]$$

$$= n\mathbb{E}_{\boldsymbol{\sigma}} \left[ \sup_{f \in \mathcal{F}_{\pm}} \sum_{i=1}^{m} 2 \left( (\sigma_{1,i} f(x_i))_+ \right)^2 \right] \ .$$

It holds

$$S^2 \leq 2nm\widehat{\mathsf{W}}_{\boldsymbol{x}}^{\mathsf{r}}(\mathcal{F}) \ . \tag{29}$$

For each $g \in \mathcal{G}$, $g(\sigma_{j,i})$ and $g(\sigma_{j',i'})$ are *independent*, though not necessarily *identically distributed*, for $(j, i) \neq (j', i')$, due to the dependence of $g(\sigma_{j,i})$ on indices $(j, i)$. It also holds, for each $g \in \mathcal{G}$, and indices $(j, i)$, that $\mathbb{E}_{\sigma_{i,j}}[g(\sigma_{i,j})] = 0$, simply due to multiplication by symmetric (Rademacher) r.v.'s.

Thus, we can use Samson's bound (Thm. 7) on $\mathcal{G}$, $\mathsf{Z}$, and $S^2$, although it is generally more convenient to work with $\mathcal{F}$ and $(\mathcal{F}_{\pm})^n$.

We now show the thesis. Fix $\varepsilon \in (0, 1)$. It follows from Samson's bound that

$$\Pr_{\boldsymbol{\sigma}} \left( \widehat{\mathsf{R}}_m(\mathcal{F}, \boldsymbol{x}) \geq \widehat{\mathsf{R}}_m^n(\mathcal{F}, \boldsymbol{x}, \boldsymbol{\sigma}) + \varepsilon \right) = \Pr_{\boldsymbol{\sigma}} \left( \mathbb{E}[\mathsf{Z}] \geq \mathsf{Z}(\boldsymbol{\sigma}) + nm\varepsilon \right) \leq \exp \left( -\frac{S^2}{4} \mathsf{h} \left( \frac{2nm\varepsilon}{S^2} \right) \right) \ .$$

The function

$$g(x) \doteq x\mathsf{h} \left( \frac{2nm\varepsilon}{x} \right)$$

is monotonically decreasing in its argument. Thus, using (29) gives

$$\Pr_{\boldsymbol{\sigma}} \left( \widehat{\mathsf{R}}_m(\mathcal{F}, \boldsymbol{x}) \geq \widehat{\mathsf{R}}_m^n(\mathcal{F}, \boldsymbol{x}, \boldsymbol{\sigma}) + \varepsilon \right) \leq \exp \left( \frac{-nm\widehat{\mathsf{W}}_{\boldsymbol{x}}^{\mathsf{r}}(\mathcal{F})}{2} \mathsf{h} \left( \frac{\varepsilon}{\widehat{\mathsf{W}}_{\boldsymbol{x}}^{\mathsf{r}}(\mathcal{F})} \right) \right) \ .$$

Now, for $u > -1/2$, define the function

$$\mathsf{h}_1(u) \doteq 1 + u - \sqrt{1 + 2u} \ .$$

Using the fact (see Boucheron et al. [7, Ch. 2.4]) that

$$\mathsf{h}(u) \geq 9\mathsf{h}_1 \left( \frac{u}{3} \right) \text{ for every } u \in (-1, +\infty),$$

we obtain

$$\Pr_{\boldsymbol{\sigma}} \left( \hat{R}_m(\mathcal{F}, \boldsymbol{x}) \geq \hat{R}_m^n(\mathcal{F}, \boldsymbol{x}, \boldsymbol{\sigma}) + \varepsilon \right) \leq \exp\left( -\frac{9}{2} nm \widehat{W}_{\boldsymbol{x}}^{\mathsf{r}}(\mathcal{F}) \mathsf{h}_1 \left( \frac{\varepsilon}{\widehat{W}_{\boldsymbol{x}}^{\mathsf{r}}(\mathcal{F})} \right) \right) \quad .$$

The result for $\hat{q}_{\mathcal{F}}(\boldsymbol{x}) = 1$ is obtained by imposing that the r.h.s. be at most $\delta$ and solving for $\varepsilon$ using standard sub-gamma inequalities. The general case then follows via linear scaling. $\quad\square$

This bound is quite comparable to Bousquet's bound on the SD (see Thm. 3). The variance factors $\widehat{W}_{\boldsymbol{x}}^{\mathsf{r}}(\mathcal{F})$ and $\widehat{W}_{\boldsymbol{x}}(\mathcal{F})$ are convenient, as they depend only on sample variances, rather than true variances and expected supremum deviations.

Even if Samson's inequality introduces additional 2-factors on both the range and variance w.r.t. Thm. 3, both are divided by MC-trial count $n$, so for $n \geq 2$ trials, the Monte-Carlo error terms become negligible.

## B  Details on the Experimental Evaluation

As mentioned in the main text, Lemma 3 is a consequence of [27, Lemmas 26.11, 26.10], reported here for completeness.[5]

**Lemma 5** (27, Lemmas 26.11, 26.10). *It holds*

$$\hat{R}_m(\mathcal{F}_1, \boldsymbol{x}) = \mathbb{E}_{\boldsymbol{\sigma}} \left[ \left\| \frac{1}{m} \sum_{i=1}^m \sigma_i x_i \right\|_\infty \right] \leq \max_i \|x_i\|_\infty \sqrt{\frac{2\ln(2d)}{m}},$$

*and*

$$\hat{R}_m(\mathcal{F}_2, \boldsymbol{x}) = \mathbb{E}_{\boldsymbol{\sigma}} \left[ \left\| \frac{1}{m} \sum_{i=1}^m \sigma_i x_i \right\|_2 \right] \leq \max_i \|x_i\|_2 \frac{1}{\sqrt{m}} \quad .$$

We now show the centralized variants.

**Lemma 3.** *Let* $\bar{x} \doteq \frac{1}{m} \sum_{i=1}^m x_i \in \mathbb{R}^d$. *For the $\ell_1$ norm, it holds*

$$\hat{R}_m(\hat{C}_{\boldsymbol{x}}(\mathcal{F}_1), \boldsymbol{x}) = \mathbb{E}_{\boldsymbol{\sigma}} \left[ \left\| \frac{1}{m} \sum_{i=1}^m \boldsymbol{\sigma}_i(x_i - \bar{x}) \right\|_\infty \right] \leq \max_i \|x_i - \bar{x}\|_\infty \sqrt{\frac{2\ln(2d)}{m}},$$

*while for the $\ell_2$ norm, it holds*

$$\hat{R}_m(\hat{C}_{\boldsymbol{x}}(\mathcal{F}_2), \boldsymbol{x}) = \mathbb{E}_{\boldsymbol{\sigma}} \left[ \left\| \frac{1}{m} \sum_{i=1}^m \boldsymbol{\sigma}_i(x_i - \bar{x}) \right\|_2 \right] \leq \max_i \|x_i - \bar{x}\|_2 \frac{1}{\sqrt{m}} \quad .$$

*Proof.* We show the $\ell_2$ case in detail; the reasoning for the $\ell_1$ case is essentially the same (see details at the end of the proof). The definition of $\hat{R}_m(\hat{C}_{\boldsymbol{x}}(\mathcal{F}_2), \boldsymbol{x})$ is

$$\hat{R}_m(\hat{C}_{\boldsymbol{x}}(\mathcal{F}_2), \boldsymbol{x}) = \mathbb{E}_{\boldsymbol{\sigma}} \left[ \sup_{w:\|w\|_2 \leq 1} \left| \frac{1}{m} \sum_{i=1}^m \sigma_i(w \cdot x_i - \hat{\mathbb{E}}_{\boldsymbol{x}}[w]) \right| \right],$$

where

$$\hat{\mathbb{E}}_{\boldsymbol{x}}[w] = \frac{1}{m} \sum_{i=1}^m (w \cdot x_i) = w \cdot \bar{x} \quad .$$

Using linearity, we then get

$$\hat{R}_m(\hat{C}_{\boldsymbol{x}}(\mathcal{F}_2), \boldsymbol{x}) = \mathbb{E}_{\boldsymbol{\sigma}} \left[ \sup_{w:\|w\|_2 \leq 1} \left| w \cdot \frac{1}{m} \sum_{i=1}^m \sigma_i(x_i - \bar{x}) \right| \right] \quad .$$

Now, for ease of notation, let $u \doteq \frac{1}{m} \sum_{i=1}^{m} \sigma_i(x_i - \bar{x})$. The supremum is realized when

$$w = \frac{u}{\|u\|_2},$$

because in this case the vector $w$ has the same direction as $u$, and the largest possible norm $\|w\|_2 = 1$. Since the two vectors $w$ and $u$ are collinear, the Cauchy-Schwarz inequality holds with equality, and we have

$$w \cdot u = \|w\|_2 \|u\|_2 = \|u\|_2 = \left\| \frac{1}{m} \sum_{i=1}^{m} \sigma_i(x_i - \bar{x}) \right\|_2 .$$

We thus obtain

$$\hat{\mathsf{R}}_m(\hat{\mathsf{C}}_{\boldsymbol{x}}(\mathcal{F}_2), \boldsymbol{x}) = \mathbb{E}_{\boldsymbol{\sigma}} \left[ \left\| \frac{1}{m} \sum_{i=1}^{m} \sigma_i(x_i - \bar{x}) \right\|_2 \right] .$$

From here, we can proceed as in the second part of the proof of [27, Lemma 26.10] to obtain the thesis.

By similar reasoning (now with Hölder's inequality in place of the Cauchy-Schwarz inequality, and following the proof of Shalev-Shwartz and Ben-David [27, Lemma 26.11]), we get that

$$\hat{\mathsf{R}}_m(\hat{\mathsf{C}}_{\boldsymbol{x}}(\mathcal{F}_1), \boldsymbol{x}) = \mathbb{E}_{\boldsymbol{\sigma}} \left[ \left\| \frac{1}{m} \sum_{i=1}^{m} \sigma_i(x_i - \bar{x}) \right\|_\infty \right] \leq \max_i \|x_i - \bar{x}\|_\infty \sqrt{\frac{2\ln(2d)}{m}} . \qquad \square$$

## B.1 Data Generation

Our data distributions for both the $\ell_1$ and $\ell_2$ constrained linear family experiments are both randomized and parameterized by dimension $d$. Rademacher averages and wimpy variances depend on the randomization and $d$, and ranges may be bounded *a priori* in terms of $d$.

$\ell_1$ **Datasets**  In our $\ell_1$ experiments, each $x_j$ is independently Beta-distributed, thus $\boldsymbol{x} \sim \mathrm{B}(\boldsymbol{\alpha}_1, \boldsymbol{\beta}_1) \times \cdots \times \mathrm{B}(\boldsymbol{\alpha}_d, \boldsymbol{\beta}_d)$. The parameters $\boldsymbol{\alpha}$ and $\boldsymbol{\beta}$ are themselves randomized, in particular, we sample $\boldsymbol{\alpha}_j$ and $\boldsymbol{\beta}_j$ from $\sqrt{\chi_j^2}$, where $\chi_k^2$ is the $\chi^2$ distribution with $k$ degrees of freedom. In these datasets, $r = q = 1$.

$\ell_2$ **Datasets**  In our $\ell_2$ experiments, we generate random *mean vector* $\boldsymbol{\mu} \in \mathbb{R}^d$ and *covariance matrix* $\boldsymbol{\Sigma} \in \mathbb{R}^{d \times d}$, then sample $\boldsymbol{x}' \sim \mathcal{N}(\mu, \boldsymbol{\Sigma})$, and finally obtain sample $\boldsymbol{x}$ by projecting $\boldsymbol{x}'$ to the nonnegative hyperquadrant of the radius $\sqrt{d}$ $\ell_2$ sphere; i.e.,

$$\boldsymbol{x} = \operatorname*{argmin}_{\boldsymbol{x} \in \mathbb{R}^d : \|\boldsymbol{x}\|_2 \leq \sqrt{d} \wedge \boldsymbol{0} \preceq \boldsymbol{x}} \|\boldsymbol{x} - \boldsymbol{x}'\|_2 .$$

Taking $\mathrm{I}_d$ to be the identity matrix, we sample $\boldsymbol{\mu} \sim \mathcal{N}(\boldsymbol{1}, \mathrm{I}_d)$, and taking $\boldsymbol{a} \sim \mathcal{U}(0,1)^{d \times d}$, we let $\boldsymbol{\Sigma} \doteq \frac{\boldsymbol{a}\boldsymbol{a}^\top}{d} + \mathrm{I}_d$. In these datasets, $r = q = \sqrt{d}$.

## B.2 Supplementary Plots

Figure 3 shows the same results as Fig. 1 (in the main text), but without the scaling of the quantities by $\sqrt{m}$. Similarly, Fig. 4 shows the same results as Fig. 2, sans scaling by $\sqrt{m}$. Additionally, both plots also include a *McDiarmid term* $3r\sqrt{\ln \frac{1}{\eta}/2m}$, representing the *additive error* incurred bounding the SD in terms of $\hat{\mathsf{R}}_m^1(\mathcal{F}, \boldsymbol{x}, \boldsymbol{\sigma})$. We stress that this term *does not* include the MC-ERA itsef, and thus is just one summand of the total McDiarmid SD bound. Nevertheless, the McDiarmid term alone asymptotically exceeds *all other bounds* in all experiments, except for the (loose) noncentralized analytical bound of $\mathcal{F}_1$ over $\mathbb{R}^{256}$. This further reinforces the improvement of *variance-sensitive* bounds over the (range-only) McDiarmid bounds.

Figure 3: Comparison of SD bounds as functions of the sample size $m$. See the main text for an explanation of the results.

Figure 4: Comparison of SD bounds as functions of the sample size $m$. See the main text for an explanation of the results.

## Footnotes

[4]To be precise, this is an immediate consequence of the statement of [7, Thm. 2.11], through an application of the Chernoff method to the moment generating function given therein.

[5]The identities in the lemma are not reported in the original, but can be easily obtained through a slightly more refined proof than the one presented in the original. See the proof of Lemma 3 for intuition.