[Reviews · NeurIPS 2020]

Review 1

Summary and Contributions: The paper derives sample-dependent high-probability bounds on the uniform deviation for a function class. The paper achieves this by analyzing empirical Rademacher average of the empirically centralized function class, i.e., function class shifted by empirical mean. The derived bounds are sample-dependent and have the right dependence on the parameters of function class: wimpy variance and range. The paper then studies the concentration of the empirical Rademacher average to the distribution Rademacher average for the centralized version. These results are obtained by showing that the empirical RA of the empirical centralization is a self bounding function with the appropriate rates, and then using the concentration of the self-bounding functions. Update: After reading author's feedback, I maintain my score.

Strengths: I think the paper makes important theoretical contributions which are also practically important: the settings where empirical estimates for the supremum deviation of the function class are weak. This problem is primarily important in the small sample regime because the existing bounds can be loose by large multiplicative factor. The paper studies ERA of the centralized function class and shows that it also satisfies bounds similar to the normal ERA but where the raw moments are replace by wimpy variance, which can be much tighter. The paper then present results for the Monte Carlo estimates that get better with n and rely on solving the supremum problem only n times.

Weaknesses: I think the paper is a good theoretical contribution for practical problems.

Correctness: I have not checked the proofs rigorously but they seem correct.

Clarity: I think the paper is well-written and the problem statement is well-motivated.

Relation to Prior Work: The paper clearly presents its contributions in the context of the related work.

Reproducibility: Yes

Additional Feedback: Some minor comments: - Line 156: 'an' - Similar to other citations in the paper, please provide a specific reference form [24] for Eq. (2). - Shouldn't b(m) in Eq. (5) be also \Omega(1/\sqrt{m}). - Please improve the presentation of Theorem 4. I understand it is a bit complicated but currently it is very difficult to parse. - Line 198 \ell_{p/1-p}


Review 2

Summary and Contributions: Revisit Rademacher complexities (understood as Conditional Rademacher averages) in order to achieve more practical risk bounds. The key idea to to replace $\mathbb{E}f$ in $\sup_{f } \left| \sum_{i=1}^n f(X_i) - \mathbb{E}f \right|$ by $P_n f$ (the empirical mean of $f$) leading to by $\sup_{f } \left| \sum_{i=1}^n f(X_i) - P_nf \right|$, and to take (conditional) Rademacher averages. The author establish non-trivial concentration inequalities for the conditional Rademacher averages of empirically centered processes. This relies on establishing a self-bounding property which is less trivial than in the case of the classic conditional Rademacher average. The author apply theses bounds to obtain more usable uniform convergence bounds. It is not obvious to see how this could enter into a model selection method. The theoretical investigation is complemented by simulations that suggest the derived bounds are better than previously known bounds.

Strengths: - Significant and timely results - Well-written - Attractive to theoretically inclined readers

Weaknesses: - How does empirical centering interact with localization? - Does empirical centering allow to refine obtain more transparent bounds on excess risk? - I am mot too impressed by the Monte-Carlo part. - Motivating the results orally could be difficult.

Correctness: So far so good

Clarity: The paper is written in clear and simple style. Yet, it could be improved in several ways. Introduction could go straight to the point: start with centered empirical processes (3) and then develop the case for empirical centering. I would be fully convinced by empirical centralization if it was shown to interact nicely with localization, that is, would it help when dealing with the setting of Koltchinskii 2006 (Local Rademacher complexities).

Relation to Prior Work: Should mention: MR2329442 (2009h:62060) Koltchinskii, Vladimir Local Rademacher complexities and oracle inequalities in risk minimization. Ann. Statist. 34 (2006), no. 6, 2593–2656. (Reviewer: Erich Haeusler) 62G30 (62H30 68T05) MR2243881 (2007k:60057) Giné, Evarist; Koltchinskii, Vladimir Concentration inequalities and asymptotic results for ratio type empirical processes. Ann. Probab. 34 (2006), no. 3, 1143–1216. (Reviewer: Przemysław Matuła) 60E15 (60F15 60F17 62G08 68T10)

Reproducibility: Yes

Additional Feedback:


Review 3

Summary and Contributions: In this paper, the authors study the supremum deviation (SD) of empirical means by Rademacher averages for function classes with empirical centralization, where each function is shifted by its empirical average. The authors then study the bias and centralization of the empirically centralized Rademacher averages. Based on this, the authors develop a sample-dependent bound for SD in terms of wimpy empirical variance and empirical Rademacher average of empirically-centralized family. Monce Carlo estimates of the empirical Rademacher average is also analyzed.

Strengths: A nice property of these bounds is that they are sample dependent: they depend on empirical variances and empirical Rademacher averages. Therefore, these bounds can be directly calculated from sample. Furthermore, the bounds involve complexity of empirically-centralized family and wimpy variance, which can be much smaller than the non-centralized counterparts. It is also shown that these bounds have optimal dependency on wimpy variance.

Weaknesses: The authors consider the supremum of absolute deviation $\sup_f|\hat{E}_x[f]-E_D[f]|$. In statistical learning theory, it suffices to estimate $\sup_{f}[E_D[f]-\hat{E}_x[f]]$, i.e., there is no absolute value. This term can be bounded by the modified empirical Rademacher average $E_\sigma\sup_f\frac{1}{m}\sum_{i=1}^{m}\sigma_if(x_i)$ (no absolute values), which is smaller than the one considered in the paper with an absolute value. For this modified empirical Rademacher average, it is clear that it is equal to the corresponding one for empirically-centralized family, i.e., \[ \ebb_\sigma \sup_f\frac{1}{m}\sum_{i=1}^{m}\sigma_if(x_i)=\ebb_\sigma \sup_f\frac{1}{m}\sum_{i=1}^{m}\sigma_i\big(f(x_i)-\hat{E}_x[f]\big). \] That is, there is no difference whether we empirically centralize the function or not if we consider Rademacher average without absolute values. Therefore, the results in this paper may not be interesting since centralization does not help for a smaller complexity that is sufficient for generalization. The centralization in terms of expectation has been considered in the literature for both Rademacher averages and variances. This paper extends this centralization to empirical centralization. Although the derived results are nice, they may not be surprising and significant enough. Section 1, w.r.t. to should be w.r.t. to In Theorem 4, should $W(F)$ there be $\hat{W}_x(F)$? ------------------ After Author Response ------------------ Thank you for your response. I agree with the authors that the Rademacher complexity with and without empirical centralization are also different when there is no absolute value.

Correctness: The theoretical analysis seems to be correct.

Clarity: The paper is well written.

Relation to Prior Work: The difference of this work from the previous contributions is clearly illustrated.

Reproducibility: Yes

Additional Feedback:


Review 4

Summary and Contributions: In this paper, the authors introduce the use of empirical centralization to derive novel practical, probabilistic, sample-dependent bounds to the Supremum Deviation (SD). Their bounds have optimal dependence on the wimpy variance and the function ranges, and the same dependence on the number of samples as existing SD bounds. From a practical standpoint, they develop tightly-concentrated Monte Carlo estimators of the empirical Rademacher average of the empirically-centralized family, and show concentration results for the empirical wimpy variance. Finally, they perform experiments showing that the bounds greatly outperform non-centralized bounds and are practical even at small sample sizes.

Strengths: The paper brings to light several theoretical bounds with seemingly sound proofs (I unfortunately was not able to go through it in extreme detail, but the structure seemed correct). The empirical evaluation was clean concise and to the point. The bounds seem novel.

Weaknesses: Although the paper is highly theoretical, a bit more exposure to some concrete applications can help in the overall readability. It would also make it more relevant to the NeurIPS community. One concrete example on how such bounds can be used in a practical application can make the paper more accessible to a lot more people. --------------------------------------------------------------------------------------------- EDIT: I thank the authors for their response, which have addressed my questions. I leave my evaluation unchanged.

Correctness: I was not able to go through the proofs in detail, but the structure looked ok.

Clarity: The writing can be improved a bit. There are a couple of typos here and there. For example: 1. Line 20: {1,-1} instead of {-1,-1} 2. Line 268: Through Some of the theoretical bounds can be simplified by using O() notation, and can help the readers as well. For example the result in Theorem 1.

Relation to Prior Work: The paper covers the related section well by clearly highlighting how their paper differ.

Reproducibility: Yes

Additional Feedback:

[Author Response · NeurIPS 2020]

We thank all the Reviewers for their feedback and their service to the community. We are glad that you have
all understood the relevance of our work and, in general, appreciated it. In the following, we try to comment
on all the raised issues, and how we will address them, which will certainly improve our work.

**Localization** We agree that the relationship between empirical centralization and localization is important
and we gave it much thought. We explained this relationship in an earlier submission, but had to omit it in
the NeurIPS submission due to space limitations. We summarize it in the following paragraphs.

Empirical centralization is *complementary and orthogonal with localization*, as it fixes a different issue:
localization is akin to a second-moment normalization (i.e., dividing out variance), whereas centralization is a
(complementary) first-moment normalization technique. Because both localized and centralized Rademacher
averages are themselves Rademacher averages, they are mutually compatible. We recommend centralizing
before localizing, as this approach addresses several issues with standard localization methods.

In particular, taking localization to mean "analysis of the variance-constrained star-convex hull", function
families containing $\{x \mapsto 0, x \mapsto 1\}$ (for example, any case of realizable symmetric classification with 0–1 loss
must include these two functions) suffer from $\Omega(1/\sqrt{m})$ convergence rates, since constant functions have
variance 0, whereas with centralization, these two functions are the same, removing this lower bound.

Instead taking localization to mean "constrain by raw-variance," the above issue is solved, but now the
localized Rademacher average bounds are $\Omega(V/\sqrt{m})$, where $V$ is still only the smallest *raw*-variance.

By centralizing before localizing, the difference vanishes, as with mean 0, raw and centralized variances
coincide. Furthermore, both of these bottlenecks are resolved: with empirical centralization, in many instances
we reduce the asymptotic gap between upper and lower SD bounds, and in this sense, our SD, and thus
excess risk, bounds are more transparent. Thus centralization and localization are complementary, and akin
to first and second moment normalizations. We intend to fully explore this connection in future work.

**Monte-Carlo approach** As mentioned on line 183, a consequence of Theorem 5 is that $n = 1$ trials are
sufficient to asymptotically match the rate for the SD shown by Bousquet (2002) (reported in Theorem 3),
whereas a use of McDiarmid's inequality (as in the state of the art w.r.t. Monte-Carlo Rademacher Averages
(Bartlett and Mendelson 2002)) would require high $n$ for low wimpy variance. Matching Bousquet's rate only
occurs with centralization: non-centralized Rademacher averages have inferior Monte-Carlo concentration
properties. We have additional experimental results showing a comparison between our results and the one
using McDiarmid's inequality, which we did not include due space limitations but we will include in the
updated supplementary materials.

**RA without absolute values** The identity

$$\mathbb{E}_\sigma \sup_f \frac{1}{m} \sum_{i=1}^m \sigma_i f(x_i) = \mathbb{E}_\sigma \sup_f \frac{1}{m} \sum_{i=1}^m \sigma_i (f(x_i) - \hat{\mathbb{E}}_x[f])$$

does not hold in general. It does hold by linearity when all functions in the family have equal expectation, but
consider the counterexample family $\{x \mapsto -1, x \mapsto 1\}$: any centralized Rademacher average (with or without
absolute values) is then 0, whereas without centralization, the value is $\Theta(1/\sqrt{m})$ (see Eq. 5). Other less
trivial counterexamples are possible but too convoluted to be explained in the limited space of this response.

Thus even the notion of Rademacher averages without absolute values benefits from centralization. Further-
more because the 2-sided symmetrization inequality (with centralization and absolute values, eq. 4) is factor-4
sharp, and by applying the non-absolute symmetrization inequality once in both directions, we recover a
2-sided guarantee, we believe that any gains obtained by removing the absolute value are marginal.

We also have evidence that, when using Rademacher averages for non-statistical-learning-theory tasks, such
as approximation algorithms for data analytics tasks (e.g., references [17,18,20,21]), the absolute value is
important, and should be considered as early as possible.

We omitted a discussion of non-absolute Rademacher averages for clarity of presentation, but we could include
this information if it is deemed sufficiently important.

**Other comments** As requested by Reviewer 4, we will give additional details about applications; in particular,
we will describe how our linear family bounds can be applied to get sharper bounds for selecting the optimal
expert in a batch-learning panel-of-experts setting (and thus reduced regret in online settings). Reviewer 2
asks how our methods extend to model selection, and we note that, for example, this panel-of-experts setting
can be extended to *structural risk minimization* if the experts are organized into concentric groups. We will
use Big-Oh notation whenever possible, and clarify the statement of Thm. 4. We will fix the typos pointed
out by the Reviewers and do a deep editing pass for any other spelling, grammar, or syntax issues.

[Meta-Review · NeurIPS 2020]

The reviewers all agree that this work makes a valuable contribution to the literature on uniform concentration bounds, and all recommend acceptance. The paper studies an alternative approach to concentration via shifting functions by their empirical mean (whereas prior work had considered shifting by their true mean). This can lead to sharper data-dependent bounds.